# Don't Just Chase "Highlighted Tokens" in MLLMs: Revisiting Visual Holistic Context Retention

**Xin Zou**[1,2], **Di Lu**[1,†], **Yizhou Wang**[1], **Yibo Yan**[1,2], **Yuanhuiyi Lyu**[1,2],
**Xu Zheng**[1,3], **Linfeng Zhang**[4], **Xuming Hu**[1,2*]

[1] The Hong Kong University of Science and Technology (Guangzhou)
[2] The Hong Kong University of Science and Technology
[3] INSAIT, Sofia University "St. Kliment Ohridski"
[4] Shanghai Jiao Tong University
https://github.com/obananas/HoloV

## Abstract

Despite their powerful capabilities, Multimodal Large Language Models (MLLMs) suffer from considerable computational overhead due to their reliance on massive visual tokens. Recent studies have explored token pruning to alleviate this problem, which typically uses text-vision cross-attention or [CLS] attention to assess and discard redundant visual tokens. In this work, we identify a critical limitation of such attention-first pruning approaches, *i.e.*, they tend to preserve semantically similar tokens, resulting in pronounced performance drops under high pruning ratios. To this end, we propose HoloV, a simple yet effective, plug-and-play visual token pruning framework for efficient inference. Distinct from previous attention-first schemes, HoloV rethinks token retention from a holistic perspective. By adaptively distributing the pruning budget across different spatial crops, HoloV ensures that the retained tokens capture the global visual context rather than isolated salient features. This strategy minimizes representational collapse and maintains task-relevant information even under aggressive pruning. Experimental results demonstrate that our HoloV achieves superior performance across various tasks, MLLM architectures, and pruning ratios compared to SOTA methods. For instance, LLaVA1.5 equipped with HoloV preserves 95.8% of the original performance after pruning 88.9% of visual tokens, achieving superior efficiency-accuracy trade-offs.

## 1 Introduction

Multimodal Large Language Models (MLLMs) have demonstrated outstanding capabilities [82, 12] in tasks such as image captioning [35, 61, 14], visual question answering [24, 99, 36], and video understanding [34, 64, 79]. However, these models [43, 78, 38] typically require converting visual inputs into long sequence representations (*i.e.*, visual tokens), which increases the computational complexity and cost of inference [97], especially for high-resolution images [41] and multi-frame videos [57], where redundant visual information further exacerbates the computational overhead.

To address this challenge, researchers have introduced token pruning strategies [49, 13, 98, 87] that aim to retain the highlighted visual tokens as well as prune others for accelerating MLLM's inference. These methods typically define importance criteria for tokens, such as attention scores [13, 19] or gradient information [59, 58], to quantify the significance of visual tokens, and less important tokens are pruned during the inference phase, which balances speed and performance, but with limitations.

---

*Corresponding author, †Equal contribution

39th Conference on Neural Information Processing Systems (NeurIPS 2025).

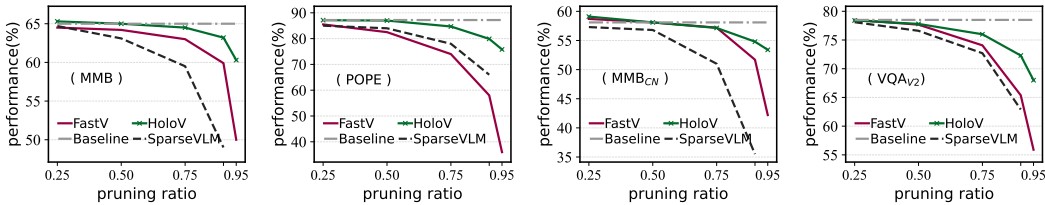

Figure 2: Relationship between performance and pruning ratios of different baseline methods. As the token pruning ratio grows, the performance of these attention-first strategies degrades dramatically, while HoloV maintains the substantial performance even at 90% and 95% of the pruning ratios.

As shown in Fig. 1, FastV [13] is an intuitive solution that ranks visual tokens based on attention distributions across different layers, and then prunes the bottom $R\%$ of tokens based on the computational budget, thus reducing visual token redundancy. Subsequently, more work has followed this paradigm [91, 98, 4], designing different strategies to prune redundant visual tokens via cross-modal (*i.e.*, text-vision) attention from LLMs. Besides, there are vision-centric pruning methods [77, 25, 94, 66, 88] (*e.g.*, FasterVLM [93]) that presume those visual tokens with low correlation to the [CLS] token in ViT [17], or those exhibit duplicated features tokens [20] to be redundant.

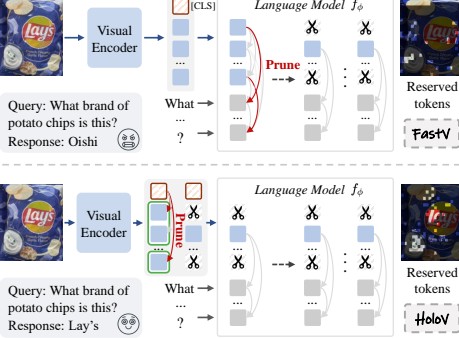

Figure 1: Snapshots of FastV and our HoloV.

Although these pruning methods can recognize the inefficiency of visual tokens in MLLMs, they are not consistently effective. As shown in Fig. 2, the performance decreases significantly as the pruning ratio increases. In our argument, this occurs because these approaches implicitly assume that *visual tokens with high attention correspond to higher informativeness*, which disregards the spatial-semantic relations of the visual scene, *i.e.*, they tend to retain tokens from localized salient regions where attention is drawn to, rather than those conducive to holistic semantic comprehension. Thus, at a high pruning ratio, such methods would only retain homologous tokens with higher scores. In a complex scene with multiple objects, retaining only "highlighted tokens" may sever relative positional and semantic connectivity information or lose key tokens associated with the subject, leading to a dramatic performance degradation. Besides, the attention mechanism introduces systematic biases [80, 81], *i.e.*, the position encoding mechanism of transformer-based MLLMs may introduce spatial priors, those in upper and lower areas visual tokens usually being assigned higher attention weights as shown in Fig. 3 right. This bias can distort the semantic contributions of the visual scene, leading the model to produce incorrect or logically contradictory inferences, or even hallucinations [100, 103]. Drawing inspiration from the above discussion, we raise the following question: *"How to locate and preserve those not highlighted but critical to visual holistic understanding tokens?"*

Cognitive science research suggests that the human visual system forms a complete semantic understanding by integrating local features with global scene cues [70, 2, 63] (*e.g.*, background textures and spatial layouts). In MLLMs, we analyzed the text-mapping relationships of different visual tokens through the strategy in [60]. As shown in Fig. 3 left, the objects in a scene could be represented by a

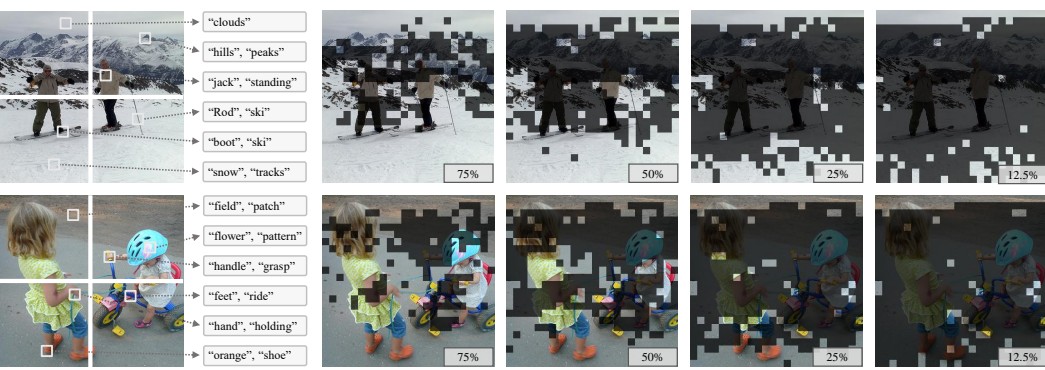

Figure 3: LEFT - Examples of textual semantics corresponding to visual tokens from scattered crops. RIGHT - Sparsification visualization examples of FastV, where retention ratios are tagged in the pics.

small number of scattered tokens, and the semantic relationships between those tokens from different regions facilitate the overall understanding, *e.g.*, *"snow", "ski", "hills"* are kind of self-explanatory. Motivated by this insight, we propose HoloV, which explicitly balances overall semantic connectivity and contextual attention during visual token pruning, addressing the critical limitation of redundancy in attention-first strategies. Our analysis demonstrates the importance of preserving visual holistic context, offering a new perspective on efficient visual token pruning in MLLMs. Through extensive experiments on diverse benchmarks and MLLM architectures, we demonstrate that HoloV consistently surpasses existing state-of-the-art token pruning approaches, achieving up to 88.9% token reduction while preserving about 96% of the original performance. Besides, HoloV is model-agnostic and easily integrable into a wide range of MLLMs, making it well-suited for practical deployment.

## 2 Related Work

### 2.1 MLLMs and Their Challenges

The recent remarkable success of Large Language Models (LLMs) [62, 95, 72, 18, 56] has spurred the trend of applying their strong capabilities to multimodal comprehension tasks, fostering the development of MLLMs [1, 69]. Leveraging open-source LLMs such as LLaMA families [72, 73, 18], MLLMs [6, 46, 47] have demonstrated enhanced adaptability across a range of visual understanding tasks, leading to a more profound ability to interpret the world. While this empowers LLMs with the capability of visual perception, the incorporation of lengthy visual tokens significantly escalates the computational burdens. Moreover, studies have shown that existing MLLMs still suffer from certain visual deficiencies [71, 32] and some hallucinations [29, 28]. Some work mitigates these issues by increasing the resolution of input images or videos [55, 86], but this further exacerbates the computational overhead. For example, LLaVA-1.5 [48] encodes a 336-resolution image into 576 visual tokens, while LLaVA-NeXT [47] doubles the resolution and generates 2,880 tokens. LLaVA-OneVision [37] represents an image using 7,290 visual tokens, and Video-LLaVA [44] faces even higher costs, as it must process numerous visual tokens from multiple frames during inference. These visual tokens occupy a large portion of the context window of their LLMs. In this work, we conducted experiments and analysis on these representative models to verify HoloV's applicability.

### 2.2 Visual Redundancy Identification

In MLLMs, visual redundancy identification facilitates the distillation of visual tokens with high informativeness for faster inference. There are two main research directions: a) Vision-centric strategies analyze the image's structure and feature distribution to discard less relevant visual tokens [13, 77]. Existing approaches include spatial-similarity clustering (*e.g.*, TokenLearner [65]), dynamic pruning based on attention scores [25, 89, 84], and using information bottleneck or entropy metrics during the prefilling stage to estimate background redundancy. b) Instruction-centric strategies typically use cross-modal attention analysis or gradient accumulation to identify redundant tokens [49, 101, 68]. Tokens with low attention or negligible gradient impact are deemed redundant [26]. Building on this, some studies explore learned importance scoring, training a lightweight end-to-end model to predict each patch's "instruction relevance," enabling even finer-grained pruning [31, 75, 91]. As the existence of language bias in LLM may cause hallucinations, we use a vision-centric scheme.

### 2.3 Visual Token Compression and Pruning

The inclusion of visual information in MLLMs introduces long token sequences, leading to high computation and memory costs. For example, mini-Gemini-HD [41] generates 2880 tokens from high-definition images, creating inference bottlenecks. To address this, research has focused on token compression and pruning techniques in Vision Transformers [10] and MLLMs [27]. Methods like LLaMA-VID [40] and DeCo [90] address this by modifying models and adding training, which increases computational costs. ToMe [11] reduces tokens without training but disrupts early cross-modal interactions [83]. LLaVA-PruMerge [66] selectively retains key tokens while merging less critical ones based on key similarity. FasterVLM [93] utilizes [CLS] attention scores from the visual encoder to re-rank and retain top visual tokens. FastV [13] and SparseVLM [98] focus on token selection using attention scores or cross-modal guidance, but overlook the role of token duplication and lack Flash-Attention [16, 15]. Our proposed HoloV maintains hard acceleration compatibility (*e.g.*, Flash-Attention), and effectively retains visual holistic context during aggressive pruning.

# 3 Preliminary and Motivation

## 3.1 Preliminary

**Architecture of MLLMs**. Given an MLLM $\mathcal{M}_\theta^{\text{MLLM}}$ parameterized by $\theta$, with a general architecture consisting of a text embedding layer, a vision encoder, a vision-text interface module, a text decoder consisting of $L$ number of transformer layers, and an affine layer which predicts the distribution of the next token. For an image-grounded text generation task, given a textual query $x$ and an input image $v$, $\mathcal{M}_\theta^{\text{MLLM}}$ first extracts vision features of $v$ by the vision encoder, and then converts them into visual tokens $z_v$ by MLP or Q-Former [76] modules. Aligned vision tokens $z_v$ are concatenated with the query $x$ as input to the text decoder, and finally decoded into a textual response $y$ autoregressive, which is formulated as: $y_t \sim p_\theta(\cdot|v, x, y_{<t}) \propto softmax(f_\theta(\cdot|v, x, y_{<t}))$, where $y_t$ indicates the $t^{th}$ token, $y_{<t}$ is the token sequence generated up to the time step $t$, and $f_\theta$ is the logit distribution.

**Attention mechanism**. Considering the computational burden associated with the length of visual tokens in MLLMs, many studies have followed the paradigm of using attention scores to evaluate the redundancy of visual tokens. Specifically, transformer-based MLLMs typically utilize causal self-attention [5] to perform computation as: Self-attention$(\mathbf{Q}, \mathbf{K}, \mathbf{V}) = \text{softmax}\left(\mathbf{Q} \cdot \mathbf{K}^\top / \sqrt{d_k}\right) \cdot \mathbf{V}$, where $d_k$ is the dimension of $\mathbf{K}$, the result of $\text{softmax}\left(\mathbf{Q} \cdot \mathbf{K}^\top / \sqrt{d_k}\right)$ is known as the attention matrix. In this work, we focus on the attention received by visual tokens from the visual [CLS] token.

## 3.2 Information Redundancy in Highlighted Tokens

When token selection is based exclusively on attention scores, the model tends to retain similar clusters, resulting in information redundancy. As shown in Fig. 4 left, adjacent tokens with similar visual features frequently receive comparable attention scores, especially in regions characterized by flat backgrounds or repetitive textures. Their spatial proximity leads these tokens to capture overlapping features, making it hard to distinguish those not highlighted yet informative tokens.

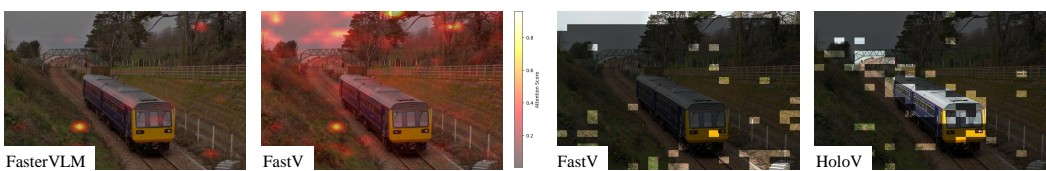

FasterVLM          FastV          FastV          HoloV

Figure 4: LEFT - Distribution map of visual token attention. RIGHT - Visualization cases of FastV and HoloV. HoloV retains contextual tokens with rich semantics, while FastV contains much redundancy.

**Positional Bias**. To further investigate attention-based token pruning methods, we take FastV as an example and visualize the distribution of the retained visual tokens. As illustrated in Fig. 4 right, the attention scores for image tokens present a consistent pattern: tokens located at the beginning and end of the sequence tend to have higher attention and are thus more likely to be preserved during pruning, leading to a positional bias. We extend our analysis by conducting statistics on samples from the text-based VQA task using the VQA V2 [23] dataset. Notably, even though these samples originate from a different task, the attention distributions of image tokens at the same layer remain highly similar, revealing recurring patterns. While the overall shape of the distributions varies slightly across layers, the set of tokens receiving relatively high attention remains stable. We suggest that this phenomenon occurs because all visual tokens are processed with text tokens in the same manner during decoding, leading to positional bias of text shift to the visual modality, *e.g.*, boundary positions of text usually imply important information, but for images, targets are mostly located in the center.

**Attention Dispersion**. In addition to positional bias, we further analyze the phenomenon of attention dispersion, i.e., a small subset of similar tokens receives the majority of attention, while most tokens are assigned low attention scores [93]. Specifically, we compute the cumulative distribution of visual tokens sorted by their attention scores, as shown in Fig. 5. The curves of last-token attention [13] and equi last attn with identical position embedding are noticeably less steep than that for [CLS] attention. It is evident that compared to [CLS] attention, text-vision attention tends to be dispersed over more visual tokens, *e.g.*, the top 20% of visual tokens account for only 40% of the total attention.

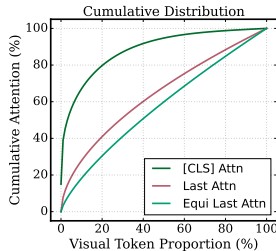

Figure 5: Cumulative distribution of different attentions.

## 3.3 Holistic Context Trumps Local Duplicates

Based on our previous analysis, attention-first token pruning methods suffer from over-localization due to positional bias and attention dispersion, *i.e.*, over-reliance on attention scores disrupts spatial-semantic relationships, *e.g.*, breaking occlusion hierarchies in multi-object interactions. Thus, our key insight is that visual token importance should be evaluated through global contextual cohesion, *i.e.*, jointly considers holistic context and local saliency rather than isolated attention magnitudes.

To further validate our hypothesis, we devised a straightforward holistic context retention strategy, *i.e.*, pruning visual tokens through random masks to retain visual information from different regions. As shown in Fig. 6 up, compared with FastV, this random strategy outperforms on more than half of the benchmarks, which demonstrates the significance of preserving holistic context for visual understanding. On the VQA text dataset, however, the random strategy failed, possibly because random pruning discards some salient fine-grained information. This result also suggests that local saliency is indispensable, especially for densely packed elements within small regions.

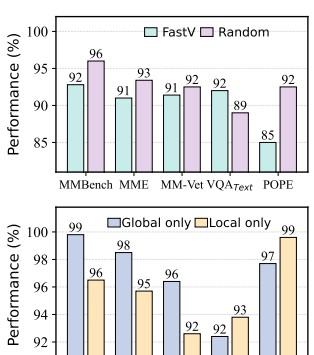

Figure 6: Up - FastV v.s. Random strategy. Down - Performance comparison of the thumbnail and local crops as inputs.

In addition, we conducted an exploratory experiment to investigate how holistic context contributes to visual understanding in MLLMs. Specifically, we use the global thumbnail and multiple local crops as visual input separately [47], and evaluate performance on the two settings against various benchmarks. As shown in Fig. 6 down, with only the global thumbnail yields strong results on general visual perception benchmarks such as MMBench [53], MME [21], and MM-Vet [92], highlighting the inherent role of holistic context in guiding general visual understanding. On the contrary, using only local crops leads to poor performance in these general perception tasks but excels in fine-grained perception benchmarks such as TextVQA [67] and POPE [42], which suggests that local duplicated saliency can offer fine-grained visual information for semantic understanding.

## 4 Methodology

Building on the above analysis, we propose HoloV, which better preserves the holistic context of images for visual understanding. By removing redundant visual tokens before the LLM decoder, our approach could make MLLMs inference faster than methods that prune tokens within the LLM. An overview of our approach is depicted in Fig. 7. In what follows, we elaborate on how our HoloV guides overall visual token compression under a high pruning ratio to keep semantic completeness.

### 4.1 HoloV Framework

To address the pivotal question raised in Sec. 1 for effective and efficient visual token pruning, we propose HoloV framework, which leverages crop-wise adaptive allocation to decentralize attention over those non-highlighted but heterogeneous tokens. Fig. 7 illustrates the core idea of HoloV.

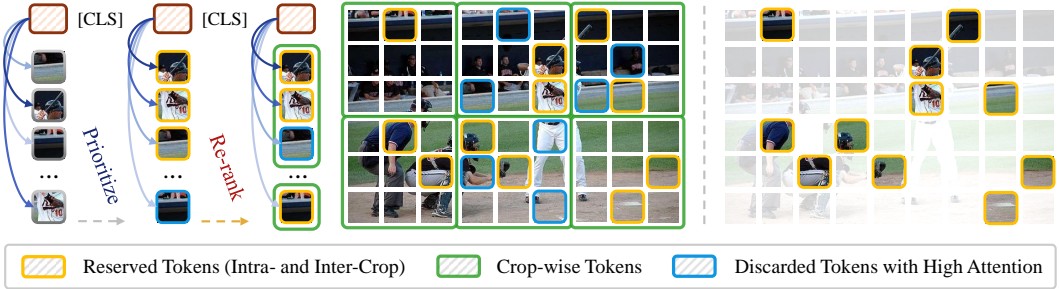

| Reserved Tokens (Intra- and Inter-Crop) | Crop-wise Tokens | Discarded Tokens with High Attention |

Figure 7: Illustration of HoloV. We re-rank highlighted visual tokens for holistic context retention.

Based on our findings about the positional bias, We first rearrange visual tokens into local crops. Let the total number of image tokens be $N_v$, which is evenly partitioned into $\mathcal{C}$ crops. This enables the model to maintain spatial granularity and gather statistics both locally and globally. Given the

normalized embeddings $\mathbf{Z}_v^c \in \mathbb{R}^{M \times d}$ in $c$-th crop, we first compute intra-crop similarity matrix $\mathbf{S}^c$ as

$$\mathbf{S}^c = (\mathbf{1} - \mathbf{I}_M) \odot \mathbf{Z}_v^c \mathbf{Z}_v^{c\top}, \tag{1}$$

where $\odot$ denotes Hadamard product, and $\mathbf{I}_M$ is the identity matrix masking self-similarities. Then, we capture intra-crop diversity by the variance of semantic distribution, the formula is as follows

$$\mathcal{V}_i^c = \frac{1}{M-1} \sum \left(\mathbf{S}_{i,j}^c - \mu_i^c\right)^2, \tag{2}$$

where a high value of $\mathcal{V}_i^c$ indicates that $i$-th token has diverse connections with others, the visual semantics expressed by the informative token is essential within the crop. To obtain holistic attention, we establish a balanced scoring mechanism combining contextual diversity and attention saliency. Specifically, we merge variance $\mathcal{V}^c$ and [CLS] attention $\mathcal{A}^c$ in the crop using adaptive scaling:

$$\mathcal{H}^c = \gamma_c \mathcal{V}^c + \mathcal{A}^c, \text{ where } \gamma_c = \mathbb{E}[\|\mathcal{A}^c\|]/\mathbb{E}[\|\mathcal{V}^c\|]. \tag{3}$$

**Adaptive holistic token allocation.** To preserve overall scene semantics and spatial diversity, we compute a crop-level priority score by averaging token scores within each crop. The total quota for selected image tokens $T'$ is dynamically allocated to crops according to their normalized crop-level importance. The allocation to each crop is discrete and capped, ensuring spatial coverage while preventing over-concentration on specific regions. We resolve rounding and overflow through an iterative reallocation procedure, so that crops with excess quota donate surplus tokens to those with remaining capacity, according to their crop-level scores.

We compute crop importance weights via

$$w_c = \left(\frac{1}{M} \sum_{t=1}^{M} \mathcal{H}_t^c\right)^\tau / \sum_{c'=1}^{\mathcal{C}} \left(\frac{1}{M} \sum_{t=1}^{M} \mathcal{H}_t^{c'}\right)^\tau, \tag{4}$$

where $\tau$ controls the sharpness of allocation. Thus, initial quota $q_c = \lfloor w_c \hat{N}_v \rfloor$, where $\hat{N}_v$ denotes the number of retained tokens. When the allocated tokens overflow or fall short, we redistribute residual tokens. For overflow, the quota is changed by $q_c = \min(q_c + \Delta_c, M), \Delta_c \propto w_c \cdot (M - q_c)$, while for fall short, we allocate the remaining quota to the crop with the highest weight. In this way, HoloV adaptively adjusts its compression degree according to the informativeness of different crops.

**Top-$k$ visual token selection.** Within each crop, select visual tokens by maximizing:

$$\operatorname{argmax}_{\Omega_c \subset \{1,\dots,M\}} \sum \mathcal{H}^c, \text{ subject to } |\Omega_c| = q_c, \tag{5}$$

which ensures both crop-wise local saliency and global relevance. We retain top-$k$ visual tokens in each crop, where $k$ is determined by the quota $q_c$ in the allocation. By performing token pruning before the LLM decoder, we dynamically adjust the number of visual tokens as input to the language model based on the actual computational budget, thus accelerating the MLLM inference.

### 4.1.1 Fast Visual Context Refetching

Motivated by the attention sinks [96], and information loss during visual token pruning, we further propose visual context refetching to fast supplement the visual holistic context. Specifically, we treat pruned tokens as supplementary evidence, re-injecting them into the MLLM through Feed Forward Network (FFN) as "key-value memory" at the middle trigger layer. This *refetch* mechanism occurs when the model exhibits high uncertainty during inference, achieving effective and efficient visual information replenishment. Limited by space, the details can be found in Appendix D.

### 4.2 Theoretical Analysis

To further justify the trustworthiness of our proposed HoloV, we provide a theoretical analysis of it. Under Assumption 1, for any pruned token, there exists a retained token that is sufficiently close in the embedding space, with bounded context variance. By leveraging the *Lipschitz continuity* [8] of the transformer layer, we can bound the semantic difference between the outputs on the original and pruned token sets. The residual error introduced by the scoring threshold is also controlled. Combining these components, we obtain the stated upper bound. More details are in Appendix C.

Table 1: Performance comparison of various methods across different benchmarks. Results are shown for different pruning ratios, with accuracy and average performance highlighted. Best results in **blue**.

| Methods | GQA | MMB | MMB$_{CN}$ | MME | POPE | SQA | VQA$_{V2}$ | VQA$_{Text}$ | VizWiz | Average |
|---|---|---|---|---|---|---|---|---|---|---|
| Upper Bound, 576 Tokens | 61.9 | 64.7 | 58.1 | 1862 | 85.9 | 69.5 | 78.4 | 58.2 | 50.0 | 100% |
| LLaVA-1.5 7B | | | | | *Retain 192 Tokens* (↓ **66.7**%) | | | | | |
| ToMe (ICLR23) | 54.3 | 60.5 | - | 1563 | 72.4 | 65.2 | 68.0 | 52.1 | - | 88.5% |
| FastV (ECCV24) | 52.7 | 61.2 | 57.0 | 1612 | 64.8 | 67.3 | 67.1 | 52.5 | 50.8 | 90.5% |
| MustDrop (2024.11) | 58.2 | 62.3 | 55.8 | 1787 | 82.6 | 69.2 | 76.0 | 56.5 | 51.4 | 97.2% |
| LLaVA-PruMerge (ICCV25) | 54.3 | 59.6 | 52.9 | 1632 | 71.3 | 67.9 | 70.6 | 54.3 | 50.1 | 91.4% |
| PDrop (CVPR25) | 57.1 | 63.2 | 56.8 | 1766 | 82.3 | 68.8 | 75.1 | 56.1 | 51.1 | 96.7% |
| FiCoCo-V (2025.03) | 58.5 | 62.3 | 55.3 | 1732 | 82.5 | 67.8 | 74.4 | 55.7 | 51.0 | 96.1% |
| HiRED (AAAI25) | 58.7 | 62.8 | 54.7 | 1737 | 82.8 | 68.4 | 74.9 | 47.4 | 50.1 | 94.6% |
| VisionZip (CVPR25) | **59.3** | 64.5 | 57.3 | 1767 | 86.4 | 68.9 | **76.8** | 57.3 | **51.6** | 98.1% |
| SparseVLM (ICML25) | 57.6 | 62.5 | 53.7 | 1721 | 83.6 | 69.1 | 75.6 | 56.1 | 50.5 | 96.1% |
| DART (EMNLP25) | 58.9 | 63.6 | 57.0 | **1856** | 82.8 | 69.8 | 76.7 | 57.4 | 51.1 | 98.5% |
| HoloV (Ours) | 59.0 | **65.4** | **58.0** | 1820 | 85.6 | 69.8 | 76.7 | **57.4** | 50.9 | **99.2%** |
| LLaVA-1.5 7B | | | | | *Retain 128 Tokens* (↓ **77.8**%) | | | | | |
| ToMe (ICLR23) | 52.4 | 53.3 | - | 1343 | 62.8 | 59.6 | 63.0 | 49.1 | - | 80.4% |
| FastV (ECCV24) | 49.6 | 56.1 | 56.4 | 1490 | 59.6 | 60.2 | 61.8 | 50.6 | 51.3 | 85.4% |
| MustDrop (2024.11) | 56.9 | 61.1 | 55.2 | 1745 | 78.7 | 68.5 | 74.6 | 56.3 | 52.1 | 95.7% |
| LLaVA-PruMerge (ICCV25) | 53.3 | 58.1 | 51.7 | 1554 | 67.2 | 67.1 | 68.8 | 54.3 | 50.3 | 89.4% |
| PDrop (CVPR25) | 56.0 | 61.1 | 56.6 | 1644 | 82.3 | 68.3 | 72.9 | 55.1 | 51.0 | 94.9% |
| FiCoCo-V (2025.03) | 57.6 | 61.1 | 54.3 | 1711 | 82.2 | 68.3 | 73.1 | 55.6 | 49.4 | 94.9% |
| HiRED (AAAI25) | 57.2 | 61.5 | 53.6 | 1710 | 79.8 | 68.1 | 73.4 | 46.1 | 51.3 | 93.1% |
| VisionZip (CVPR25) | 57.6 | 63.4 | 56.7 | 1768 | 84.7 | 68.8 | 75.6 | 56.8 | 52.0 | 97.2% |
| SparseVLM (ICML25) | 56.0 | 60.0 | 51.1 | 1696 | 80.5 | 67.1 | 73.8 | 54.9 | 51.4 | 93.8% |
| DART (EMNLP25) | **57.9** | 63.2 | **57.0** | **1845** | 80.1 | 69.1 | **75.9** | 56.4 | 51.7 | 97.5% |
| HoloV (Ours) | 57.7 | **63.9** | 56.5 | 1802 | 84.0 | 69.8 | 75.5 | 56.8 | 51.5 | **98.0%** |
| LLaVA-1.5 7B | | | | | *Retain 64 Tokens* (↓ **88.9**%) | | | | | |
| ToMe (ICLR23) | 48.6 | 43.7 | - | 1138 | 52.5 | 50.0 | 57.1 | 45.3 | - | 70.1% |
| FastV (ECCV24) | 46.1 | 48.0 | 52.7 | 1256 | 48.0 | 51.1 | 55.0 | 47.8 | 50.8 | 76.7% |
| MustDrop (2024.11) | 53.1 | 60.0 | 53.1 | 1612 | 68.0 | 63.4 | 69.3 | 54.2 | 51.2 | 90.1% |
| LLaVA-PruMerge (ICCV25) | 51.9 | 55.3 | 49.1 | 1549 | 65.3 | 68.1 | 67.4 | 54.0 | 50.7 | 87.7% |
| PDrop (CVPR25) | 41.9 | 33.3 | 50.5 | 1092 | 55.9 | 68.6 | 69.2 | 45.9 | 50.7 | 77.5% |
| FiCoCo-V (2025.03) | 52.4 | 60.3 | 53.0 | 1591 | 76.0 | 68.1 | 71.3 | 53.6 | 49.8 | 91.5% |
| HiRED (AAAI25) | 54.6 | 60.2 | 51.4 | 1599 | 73.6 | 68.2 | 69.7 | 44.2 | 50.2 | 89.4% |
| VisionZip (CVPR25) | 55.1 | 60.1 | **55.4** | 1690 | 77.0 | 69.0 | 72.4 | **55.5** | **52.9** | 94.5% |
| SparseVLM (ICML25) | 52.7 | 56.2 | 46.1 | 1505 | 75.1 | 62.2 | 68.2 | 51.8 | 50.1 | 87.3% |
| DART (EMNLP25) | **55.9** | 60.6 | 53.2 | **1765** | 73.9 | **69.8** | 72.4 | 54.4 | 51.6 | 93.9% |
| HoloV (Ours) | 55.3 | **63.3** | 55.1 | 1715 | **80.3** | 69.5 | **72.8** | 55.4 | 52.8 | **95.8%** |

## 4.3 Computational Complexity

As language instructions are much shorter than visual tokens, we focus on the FLOPs contributed by visual tokens. Let $n$ denote the number of visual tokens, $d$ the hidden size, and $m$ the FFN intermediate size (with SwiGLU). For the prefill stage, the FLOPs per transformer layer can be approximated as $an^2d + bnd^2 + cndm$, where $a$, $b$, and $c$ are constants. If the token count is reduced by a ratio $R$ ($\hat{n} = (1 - R)n$), the FLOPs reduction ratio is:

$$F = 1 - \frac{a\hat{n}^2d + b\hat{n}d^2 + c\hat{n}dm}{an^2d + bnd^2 + cndm}. \tag{6}$$

For large $n$, the quadratic term dominates, so $F \approx 1 - (1 - R)^2 = 2R - R^2$. Thus, the reduction is slightly better than linear in $R$. In the decode stage (with KV cache), the complexity becomes linear in $n$, and the FLOPs per layer are $bd^2 + (bd + cdm)n$, so the reduction is nearly proportional to $R$. HoloV speeds up inference by pruning ahead of the LLM to avoid KV cache inefficiency.

## 5 Experiments

### 5.1 Experimental Setup

**Benchmarks.** We conducted experiments on several widely used visual understanding benchmarks. For image understanding task, we performed experiments on ten widely used benchmarks, including GQA [30], MMBench (MMB) and MMB-CN [53], MME [21], POPE [42], VizWiz [9], SQA (ScienceQA) [54], VQA$_{V2}$ (VQA V2) [23], VQA$_{Text}$ (TextVQA) [67], and MM-Vet [92]. Video

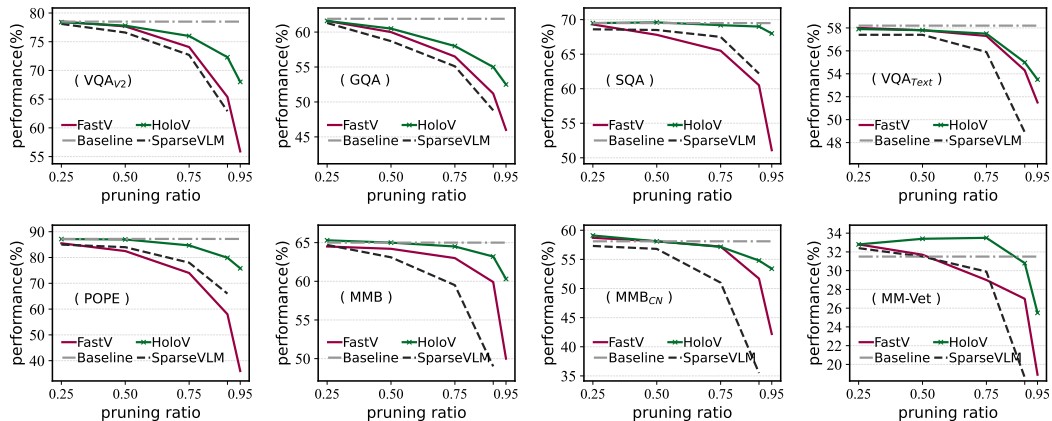

Figure 8: Comparison of different methods across multiple benchmarks under varying pruning ratios.

QA benchmarks include MSVD-QA and MSRVTT-QA [85]. All experiments on these benchmarks follow the default settings. More details of the benchmarks are provided in Appendix A.1.

**Comparison methods.** We compare our approach with several representative methods for accelerating multi-modal language models (MLLMs) via token reduction, including ToMe [11], FastV [13], SparseVLM [98], HiRED [4], LLaVA-PruMerge [66], PDrop [83], MustDrop [49], FasterVLM [93], GlobalCom$^2$[52], VisionZip [88], DART [81]. These baselines employ diverse strategies such as token merging, attention-based pruning, adaptive allocation, and hierarchical retention to improve efficiency by reducing redundant tokens. Each method offers a unique perspective on balancing computational cost and model performance. More details of baselines are provided in Appendix A.2.

## 5.2 Main Results

**General-purpose benchmarks**. We evaluate the performance of HoloV on general-purpose datasets, *i.e.*, GQA, MM-Vet, MME, MMBench, SQA, and VizWiz. As shown in Tab. 1, HoloV consistently outperforms competing approaches at different pruning ratios, *e.g.*, HoloV removes up to 88.9% of visual tokens with only a 4.2% performance drop, and 77.8% with just 2% on average.

Further, we show more results under varying pruning ratios, as shown in Fig. 8, the performance of FastV and SparseVLM drops dramatically under high pruning ratios, while HoloV maintains robust performance with relatively minor losses at all pruning ratios on SQA and MMBench. On MMBench$_{CN}$ and MM-Vet, HoloV even achieves higher than baseline (unpruned) scores at pruning ratios of 25%, 50%, and 75% (MM-Vet), then the score slowly drops as the pruning ratio increases. For VizWiz evaluation, the result in Fig. 9 indicates that HoloV can consistently obtain performance improvements at different pruning ratios, even at 95%, which means HoloV effectively retains visual holistic semantics.

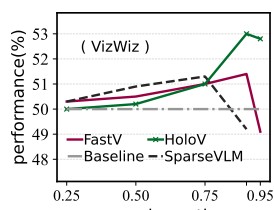

Figure 9: Performance of different methods on VizWiz under varying pruning ratios.

**Hallucination benchmarks validation**. We conduct the hallucination evaluations on POPE and MME benchmarks, with results on LLaVA-1.5-7B presented in Tab. 1, where the proposed HoloV shows robust capabilities, and the performance significantly exceeds the results of the compared SOTA methods, *e.g.*, with a pruning rate of 88.9%, HoloV achieves 80.3% accuracy compared to 76% for the second runner-up on POPE, and achieved desirable performance on MME evaluation, compared to other comparative approaches.

## 5.3 HoloV with Higher Resolution

For further comprehensive evaluation, we also evaluated HoloV for LLaVA-NeXT on different benchmarks mentioned above, with comparison to current SOTA approaches. LLaVA-NeXT introduces a new image processing method, leading to dynamic lengths of visual embeddings for various image inputs. Thus, during the evaluation, 320 visual tokens has been kept (from up to 2880 raw tokens). As shown in Table 3, the evaluation results of all various benchmarks show that HoloV obtained the highest score on almost every track, and has an average of 95. 6%, much higher than the current SOTA of 93.3%.

Table 3: Performance comparison of various methods across different benchmarks. Results are shown for different pruning ratios, with accuracy and average performance highlighted. Best results in **blue**.

| Methods | GQA | MMB | MMB$_{CN}$ | MME | POPE | SQA | VQA$_{V2}$ | VQA$_{Text}$ | VizWiz | Average |
|---|---|---|---|---|---|---|---|---|---|---|
| Upper Bound, 2880 Tokens | 64.2 | 67.4 | 60.6 | 1851 | 86.5 | 70.1 | 81.8 | 64.9 | 57.6 | 100% |
| LLaVA-NeXT 7B | | | | | *Retain 320 Tokens* (↓ **88.9%**) | | | | | |
| FastV (ECCV24) | 55.9 | 61.6 | 51.9 | 1661 | 71.7 | 62.8 | 71.9 | 55.7 | 53.1 | 88.0% |
| LLaVA-PruMerge (ICCV25) | 53.6 | 61.3 | 55.3 | 1534 | 60.8 | 66.4 | 69.7 | 50.6 | 54.0 | 85.6% |
| PDrop (CVPR25) | 56.4 | 63.4 | 56.2 | 1663 | 77.6 | 67.5 | 73.5 | 54.4 | 54.1 | 90.9% |
| MustDrop (2024.11) | 57.3 | 62.8 | 55.1 | 1641 | 82.1 | 68.0 | 73.7 | **59.9** | 54.0 | 92.2% |
| FasterVLM (ICCV25) | 56.9 | 61.6 | 53.5 | 1701 | 83.6 | 66.5 | 74.0 | 56.5 | 52.6 | 91.1% |
| HiRED (AAAI25) | 59.3 | 64.2 | 55.9 | 1690 | 83.3 | 66.7 | 75.7 | 58.8 | 54.2 | 93.3% |
| SparseVLM (ICML25) | 56.1 | 60.6 | 54.5 | 1533 | 82.4 | 66.1 | 71.5 | 58.4 | 52.0 | 89.7% |
| GlobalCom$^2$ (2025.3) | 57.1 | 61.8 | 53.4 | 1698 | 83.8 | 67.4 | 76.7 | 57.2 | 54.6 | 92.2% |
| DART (EMNLP25) | 61.7 | 65.3 | **58.2** | 1710 | **84.1** | 68.4 | 79.1 | 58.7 | **56.1** | 93.9% |
| HoloV (Ours) | **61.7** | **65.3** | 57.5 | **1738** | 83.9 | **68.9** | **79.5** | 58.7 | 55.3 | **95.6%** |

Table 4: Real inference comparison on POPE. Experiments adopt 66.7% and 90% pruning ratios.

| Methods | Time | Prefill | Latency | Mem. | Acc. | Time | Prefill | Latency | Mem. | Acc. |
|---|---|---|---|---|---|---|---|---|---|---|
| Upper Bound, 576 Tokens | 49:41 | 0.5ms | 0.334s | 19.0G | 100.% | 49:41 | 0.5ms | 0.334s | 19.0G | 100.% |
| LLaVA-1.5-7B | | *Retain 192 Tokens* (↓ **66.7%**) | | | | | *Retain 58 Tokens* (↓ **90%**) | | | |
| FastV (ECCV24) | 35:34 | 0.5ms | 0.239s | 16.0G | 75.4% | 30:41 | 0.5ms | 0.206s | 15.6G | 66.8% |
| MustDrop (2024.11) | 32:30 | 0.5ms | 0.273s | 15.6G | 96.2% | 29:40 | 0.6ms | 0.199s | 14.5G | 87.1% |
| FasterVLM (ICCV25) | **30:09** | 0.5ms | **0.202s** | 15.6G | 100.% | 25:08 | 0.5ms | **0.168s** | 14.5G | 92.5% |
| HiRED (AAAI25) | 30:08 | 0.6ms | 0.210s | 15.7G | 96.4% | **25:03** | 0.6ms | 0.168s | 14.5G | 92.7% |
| SparseVLM (ICML25) | 40:51 | 0.6ms | 0.251s | 15.8G | 97.3% | 31:28 | 0.6ms | 0.212s | 14.6G | 92.3% |
| HoloV (Ours) | 31:02 | 0.5ms | 0.208s | **15.6G** | **99.7%** | 27:36 | 0.5ms | 0.176s | **14.5G** | **95.7%** |

Besides, on video understanding benchmarks, HoloV maintains close to the original performance, significantly outperforming FasterVLM and FastV, as shown in Table 2. This demonstrates the value of HoloV when it comes to high-resolution visual input.

Table 2: Video QA Evaluations of different methods with 50% of visual tokens retained. HoloV beats SOTA.

| Methods | MSVD-QA | | MSRVT-QA | | Avgerge | |
|---|---|---|---|---|---|---|
| | Acc. | Score | Acc. | Score | Acc. | Score |
| Video-ChatGPT 7B | 64.9 | 3.3 | 49.3 | 2.8 | 57.1 | 3.1 |
| Video-LLaVA 7B | 70.2 | 3.9 | 57.3 | 3.5 | 63.8 | 3.7 |
| FastV (ECCV24) | 71.0 | 3.9 | 55.0 | 3.5 | 63.0 | 3.7 |
| FasterVLM (ICCV25) | 70.5 | 3.9 | 56.2 | 3.5 | 63.4 | 3.7 |
| DART (EMNLP25) | 71.0 | 4.0 | **56.7** | 3.6 | 58.0 | 3.7 |
| HoloV (Ours) | **71.0** | **4.0** | 56.5 | **3.6** | **63.7** | **3.7** |

## 5.4 Efficiency Analysis

To assess the efficiency of HoloV, we compare total inference time, prefill time, end-to-end latency, GPU memory usage, and accuracy on LLaVA-1.5-7B. As shown in Tab. 4, under a 90% pruning ratio, HoloV achieves a 42.7% reduction in inference time and a 42.8% decrease in latency, with only a 4.3% drop in accuracy, similarly under 66.7% pruning ratio. Compared to FastV and SparseVLM, HoloV uses less memory and runs faster. Although FasterVLM offers slightly quicker inference, HoloV improves accuracy by 3.0%, demonstrating a better balance between efficiency and performance.

## 5.5 Ablation Analysis of Crop Numbers

Partition granularity does not affect pruning efficiency: retained visual tokens are determined by pruning quotas, and the quota per crop, *i.e.*, calculated dynamically via intra-crop visual token informativeness, leaves total pruning quotas unchanged. For high-resolution images, dynamic crop number adjustment is beneficial: using fewer crops for high-detail areas and more for low-detail regions. Specifically, Table 5 shows results when total crops vary from 4 to 16, where the values represent percentages relative to original performance. We observe no significant performance impact from varying crop numbers.

Table 5: Ablation of different crop numbers.

| Methods | # = 4 | # = 8 | # = 12 | # = 16 |
|---|---|---|---|---|
| Upper Bound | 100% | 100% | 100% | 100% |
| LLaVA-1.5-7B | *Token Pruning Rate = 66.7%* | | | |
| HoloV (Ours) | 95.1% | **96.7%** | 96.1% | 94.9% |
| LLaVA-1.5-7B | *Token Pruning Rate = 77.8%* | | | |
| HoloV (Ours) | 94.5% | **95.1%** | 94.6% | 94.8% |
| LLaVA-1.5-7B | *Token Pruning Rate = 88.9%* | | | |
| HoloV (Ours) | 89.3% | 89.3% | 90.0% | **91.2%** |

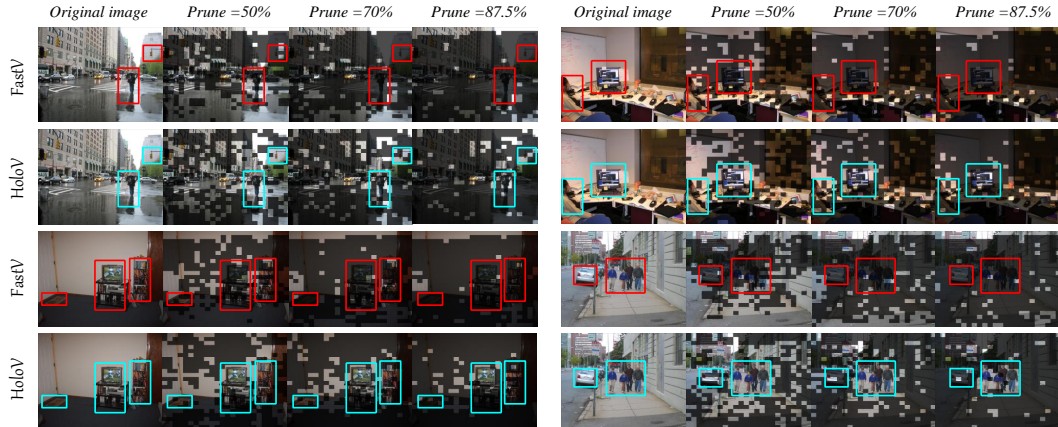

Figure 10: The case comparison between FastV and HoloV from the GQA. It presents original images alongside their pruned versions at pruning rates of 50%, 70%, and 87.5%. The bounding boxes highlight specific regions and objects across images, where HoloV well preserves the pivotal tokens.

## 5.6 Visualization Analysis

Further, we visualize retained visual patches under different pruning rates. As shown in Fig. 10, black areas indicate discarded tokens, while colored regions show key semantic areas aligned with text. Compared to FastV, HoloV preserves more relevant visual cues even under high pruning (e.g., 87.5%), effectively filtering out redundant visual tokens while keeping critical objects. This supports better cross-modal alignment, allowing pivotal holistic tokens for visual overall understanding.

## 5.7 HoloV with Qwen Architecture

To verify the architectural generalization of HoloV beyond LLaVA-based models, we conduct experiments on the Qwen2.5-VL-7B [7] architecture. As shown in Tab. 6, HoloV demonstrates strong generalization capability across this architecture, consistently outperforming the text-visual attention-based FastV at various reduction ratios, highlighting its robustness and adaptability to different model designs. Notably, it achieves average performance retention

Table 6: Comparative Experiments on Qwen2.5-VL-7B.

| Methods | MMB | MME | POPE | SQA | VQA$_{Text}$ | Avg. |
|---|---|---|---|---|---|---|
| Upper Bound | 82.8 | 2304 | 86.1 | 84.7 | 84.8 | 100% |
| Qwen2.5-VL-7B | *Token Pruning Rate = 66.7%* | | | | | |
| FastV (ECCV24) | 75.7 | 2072 | 82.2 | 78.5 | 77.9 | 92.3% |
| HoloV (Ours) | **78.3** | **2093** | **85.0** | **79.8** | **78.9** | **94.6%** |
| Qwen2.5-VL-7B | *Token Pruning Rate = 77.8%* | | | | | |
| FastV (ECCV24) | 74.9 | 2036 | 80.7 | 78.0 | 69.0 | 89.2% |
| HoloV (Ours) | **76.5** | **2043** | **82.3** | **79.8** | **70.3** | **92.7%** |
| Qwen2.5-VL-7B | *Token Pruning Rate = 88.9%* | | | | | |
| FastV (ECCV24) | 69.2 | 1940 | 78.6 | 77.4 | 60.3 | 84.3% |
| HoloV (Ours) | **72.4** | **2006** | **80.7** | **79.5** | **61.8** | **90.5%** |

rates of 94.6%, 92.7%, and 90.5% at 66.7%, 77.8%, and 88.9% token pruning rates respectively, significantly higher than FastV's 92.3%, 89.2%, and 84.3% performance. These results show that our proposed holistic pruning strategy effectively generalizes across different MLLM architectures.

## 6 Conclusion

We present HoloV, a holistic token pruning framework that addresses two critical limitations of attention-based visual compression: 1) semantic fragmentation from over-pruning non-salient regions, and 2) static importance estimation ignoring token interdependencies. The core innovation lies in variance-modulated dynamic scoring and capacity-constrained allocation, which preserve holistic context. Extensive experiments validate our method's effectiveness in maintaining both perceptual details and abstract spatial reasoning capabilities under aggressive token reduction.

## Acknowledgments and Disclosure of Funding

This work was supported by the National Natural Science Foundation of China (Grant No.62506318); Guangdong Provincial Department of Education Project (Grant No.2024KQNCX028); CAAI-Ant Group Research Fund; Scientific Research Projects for the Higher-educational Institutions (Grant No.2024312096), Education Bureau of Guangzhou Municipality; Guangzhou-HKUST(GZ) Joint Funding Program (Grant No.2025A03J3957), Education Bureau of Guangzhou Municipality.

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

# Contents of Technical Appendices

# ∞ Technical Appendices and Supplements

In this appendix, we first provide the details of the experimental setup, including information about the datasets, model architectures, and comparison methods. Then, we offer a more detailed computational complexity and theoretical analysis, along with more visualizations and insights.

## A Detailed Experiment Settings

### A.1 Benchmarks and Metrics

We conducted experiments on several widely used visual understanding benchmarks. For image understanding task, we performed experiments on ten widely used benchmarks, including GQA [30], MMBench (MMB) and MMB-CN [53], MME [21], POPE [42], VizWiz [9], SQA (ScienceQA) [54], $VQA_{V2}$ (VQA V2) [23], $VQA_{Text}$ (TextVQA) [67], and MMVet [92].

**GQA** [30] The GQA benchmark is composed of three main components: scene graphs, questions, and images. The image section encompasses not only the images themselves but also their spatial features and the attributes of all objects within the images. The questions in GQA are specifically crafted to assess the model's ability to comprehend visual scenes and engage in reasoning about different aspects of the images.

**MMBench** [53]. MMBench provides a comprehensive evaluation of a model's performance across multiple dimensions. It is structured into three levels of ability dimensions. The first level (L-1) focuses on two core abilities: perception and reasoning. Building on this foundation, the second level (L-2) includes six sub-abilities, further elaborating the model's capabilities. At the third level (L-3), the evaluation becomes more granular, encompassing 20 specific ability dimensions, thus ensuring a detailed and multi-faceted analysis of the model's performance.

**MME** [21]. The MME benchmark is another holistic evaluation framework, designed to thoroughly assess various facets of a model's performance. It includes 14 distinct subtasks, each targeting specific perceptual and cognitive abilities of the model. By employing carefully crafted instruction-answer pairs and maintaining concise instruction designs, the benchmark minimizes issues such as data leakage and unfair evaluation, ensuring a fair and reliable performance assessment.

**POPE** [42]. POPE focuses on evaluating the degree of Object Hallucination in models. It reformulates hallucination evaluation by prompting the model with specific binary questions regarding the presence of objects in images. Key metrics such as Accuracy, Recall, Precision, and F1 Score are utilized to measure the hallucination level across three different sampling strategies, providing a robust and precise evaluation of the model's object detection and hallucination behavior.

**ScienceQA** [54]. ScienceQA spans many domains, including natural sciences, language sciences, and social sciences. Questions are categorized within each domain according to topics, categories, and skills, which results in 26 topics, 127 categories, and 379 skills. This hierarchical categorization facilitates a thorough and diverse range of scientific questions, enabling an in-depth evaluation of the model's multimodal understanding, multi-step reasoning abilities, and interpretability.

**VQA-V2** [23]. VQA-V2 is designed to evaluate a model's visual perception capabilities through open-ended questions. It consists of 265,016 images representing a wide variety of real-world scenes and objects, providing rich visual contexts for the associated questions. Each question is accompanied by 10 ground truth answers provided by human annotators, enabling a comprehensive evaluation of the model's ability to answer questions accurately and effectively.

**TextVQA** [67]. TextVQA focuses on the integration of text within images, evaluating the model's ability to comprehend and reason about both the visual and textual information present. The benchmark includes a series of visual question-answering tasks where the model must not only interpret the visual content but also read and understand the embedded text in order to respond correctly.

**MMVet** [92]. MMVet is designed to assess a model's ability to solve complex tasks by leveraging various core vision-language capabilities. It defines six core vision-language capabilities and examines 16 distinct integrations of these capabilities. This allows for a nuanced evaluation of how well models integrate and utilize multiple vision-language abilities to solve tasks.

**MSVD-QA** [85]. The MSVD-QA benchmark is derived from the Microsoft Research Video Description (MSVD) dataset and consists of 1970 video clips paired with approximately 50.5K question-

answer pairs. The questions span a wide range of topics and aspects related to the video content, making it suitable for video question-answering and video captioning tasks. The questions fall into five categories: what, who, how, when, and where, providing a comprehensive set of queries for model evaluation.

**MSRVTT-QA** [85]. MSRVTT-QA includes 10,000 video clips and 243,000 question-answer pairs. One of its primary challenges lies in understanding and reasoning about video content, which involves both visual and temporal aspects. To answer questions accurately, models must effectively integrate and process these components. Similar to MSVD-QA, the tasks in MSRVTT-QA are categorized into five question types: what, who, how, when, and where, allowing for detailed performance evaluation across multiple dimensions.

## A.2  Backbones and Baselines

**Models**. We evaluate HoloV using various open-source MLLMs. For image understanding tasks, experiments are conducted on the LLaVA family, including LLaVA-1.5[2] [48] and LLaVA-NeXT[3] [47], with the latter used to validate performance on high-resolution images. For video understanding tasks, we use Video-LLaVA [43] as the baseline model. Following the settings reported in their paper.

We analyze multiple representative methods for accelerating MLLM inference through visual token pruning. These methods share the goal of improving efficiency by reducing redundant visual tokens.

**ToMe** [11] merges similar tokens in visual transformer layers through lightweight matching techniques, achieving acceleration without requiring additional training.

**LLaVA-PruMerge** [66] combines pruning and merging strategies by dynamically removing less important tokens using sparse CLS-visual attention and clustering retained tokens based on key similarity.

**FastV** [13] focuses on early-stage token pruning by leveraging attention maps, effectively reducing computational overhead in the initial layers.

**HiRED** [4] allocates token budgets across image partitions based on CLS token attention, followed by the selection of the most informative tokens within each partition, ensuring spatially aware token reduction.

**PDrop** [83] adopts a progressive token-dropping strategy across model stages, forming a pyramid-like token structure that balances efficiency and performance.

**FasterVLM** [93] evaluates token importance via CLS attention in the encoder and performs pruning before interaction with the language model, streamlining the overall process.

**MustDrop** [49] integrates multiple strategies, including spatial merging, text-guided pruning, and output-aware cache policies, to reduce tokens across various stages.

**GlobalCom**$^2$ [52] introduces a hierarchical approach by coordinating thumbnail tokens to allocate retention ratios for high-resolution crops while preserving local details.

**SparseVLM** [98] ranks token importance using cross-modal attention and introduces adaptive sparsity ratios, complemented by a novel token recycling mechanism.

**VisionZip** [88] evaluates token importance via attention in the encoder and clustering retained tokens based on key similarity.

**DART** [81] introduces a duplication-aware token reduction method that selects a small subset of pivot tokens, calculates cosine similarity between pivot tokens and remaining tokens, retains those with the lowest duplication to pivots, achieving significant acceleration while maintaining performance and good compatibility with efficient attention operators. These methods collectively highlight diverse approaches to token reduction, ranging from attention-based pruning to adaptive merging, offering complementary solutions for accelerating MLLMs.

---

[2]https://huggingface.co/liuhaotian/llava-v1.5-7b
[3]https://huggingface.co/liuhaotian/llava-v1.6-vicuna-7b

Table 7: Fine-grained comparison MMBench [53] between FastV and HoloV at high pruning ratios.

| Category (dev) | Vanilla (576 Tokens) | FastV ↓ **90%** (58 Tokens) | HoloV ↓ **90%** (58 Tokens) | FastV ↓ **75%** (144 Tokens) | HoloV ↓ **75%** (144 Tokens) |
|---|---|---|---|---|---|
| Action Recognition | 90.7 | 85.2 | 85.3 | 87.0 | 89.7 |
| Attribute Comparison | 50.0 | 50.0 | 53.9 | 52.3 | 48.7 |
| Attribute Recognition | 79.7 | 68.9 | 71.7 | 77.0 | 79.7 |
| Celebrity Recognition | 79.8 | 76.8 | 74.7 | 78.8 | 78.8 |
| Function Reasoning | 75.9 | 72.2 | 83.9 | 75.9 | 83.9 |
| Future Prediction | 45.0 | 30.0 | 58.3 | 40.0 | 58.3 |
| Identity Reasoning | 93.3 | 86.7 | 97.5 | 95.6 | 97.7 |
| Image Emotion | 78.0 | 58.0 | 68.7 | 78.0 | 76.0 |
| Image Quality | 35.8 | 22.6 | 38.8 | 28.3 | 40.1 |
| Image Scene | 96.2 | 90.4 | 91.5 | 96.2 | 97.1 |
| Image Style | 77.4 | 73.6 | 71.7 | 77.4 | 77.4 |
| Image Topic | 83.3 | 80.6 | 92.9 | 83.3 | 83.3 |
| Nature Relation | 41.7 | 39.6 | 49.4 | 37.5 | 37.5 |
| Object Localization | 39.5 | 35.8 | 23.3 | 37.0 | 38.3 |
| OCR | 59.0 | 59.0 | 81.8 | 59.0 | 84.4 |
| Physical Property Reasoning | 50.7 | 60.3 | 49.3 | 53.3 | 58.0 |
| Physical Relation | 33.3 | 41.7 | 32.7 | 41.7 | 41.7 |
| Social Relation | 88.4 | 53.5 | 75.8 | 72.1 | 75.7 |
| Spatial Relationship | 17.8 | 17.8 | 18.5 | 17.8 | 18.5 |
| Structured Image-Text Understanding | 26.9 | 30.8 | 21.8 | 28.2 | 21.9 |

## A.3 Reproducibility

**Implementaion Details**. All of our experiments are conducted on Nvidia A800-80G GPU. The implementation was carried out in Python 3.10, utilizing PyTorch 2.1.2, and CUDA 11.8. All baseline settings follow the original paper. We set $num_{crop} = \lceil 1024/N \rceil$, where $N$ denotes the number of retained visual tokens, thus the smaller the quota, the more crops there will be for visual holistic context retention.

## B More Sparsification Visualization

We conduct a detailed visualization of retained visual patches across varying pruning rates to illustrate the effectiveness of HoloV. As depicted in Fig. 11, 12, 13, the black regions represent discarded visual tokens, whereas the colored areas highlight key semantic zones that align with textual descriptions, demonstrating how HoloV strategically preserves informative content. Compared to FastV, a representative attention-based pruning method, HoloV exhibits superior capability in retaining relevant visual cues even at extremely high pruning ratios, such as 87.5%. This is achieved through its holistic pruning strategy, which prioritizes spatial-semantic diversity over isolated attention scores. By dynamically allocating pruning budgets across different image crops, HoloV effectively filters out redundant tokens while safeguarding critical objects and their contextual relationships. For instance, in complex scenes with multiple interacting elements, HoloV ensures that tokens corresponding to both focal objects and their surrounding environmental cues are preserved, whereas FastV tends to over-concentrate on high-attention regions, leading to loss of contextual coherence. This enhanced preservation of visual holistic understanding facilitates more accurate cross-modal alignment between visual features and language tokens, enabling MLLMs to maintain robust semantic reasoning capabilities even under aggressive token reduction. The visualization not only validates the superiority of HoloV's design philosophy but also provides empirical evidence of its ability to balance efficiency and semantic integrity in visual token pruning.

## B.1 MMBench Finegrained Results

As shown in Table 7, in the MMBench [53] fine-grained comparison between FastV [13] and HoloV at 90% and 75% pruning ratios, significant performance improvements are evident with HoloV in several categories. Specifically, HoloV shows enhanced outcomes in Action Recognition, Attribute Recognition, Future Prediction, Identity Reasoning, Image Emotion, Image Quality, and Image Scene. These results underline HoloV's ability to retain crucial visual information for complex understanding and response capabilities within dynamic environments.

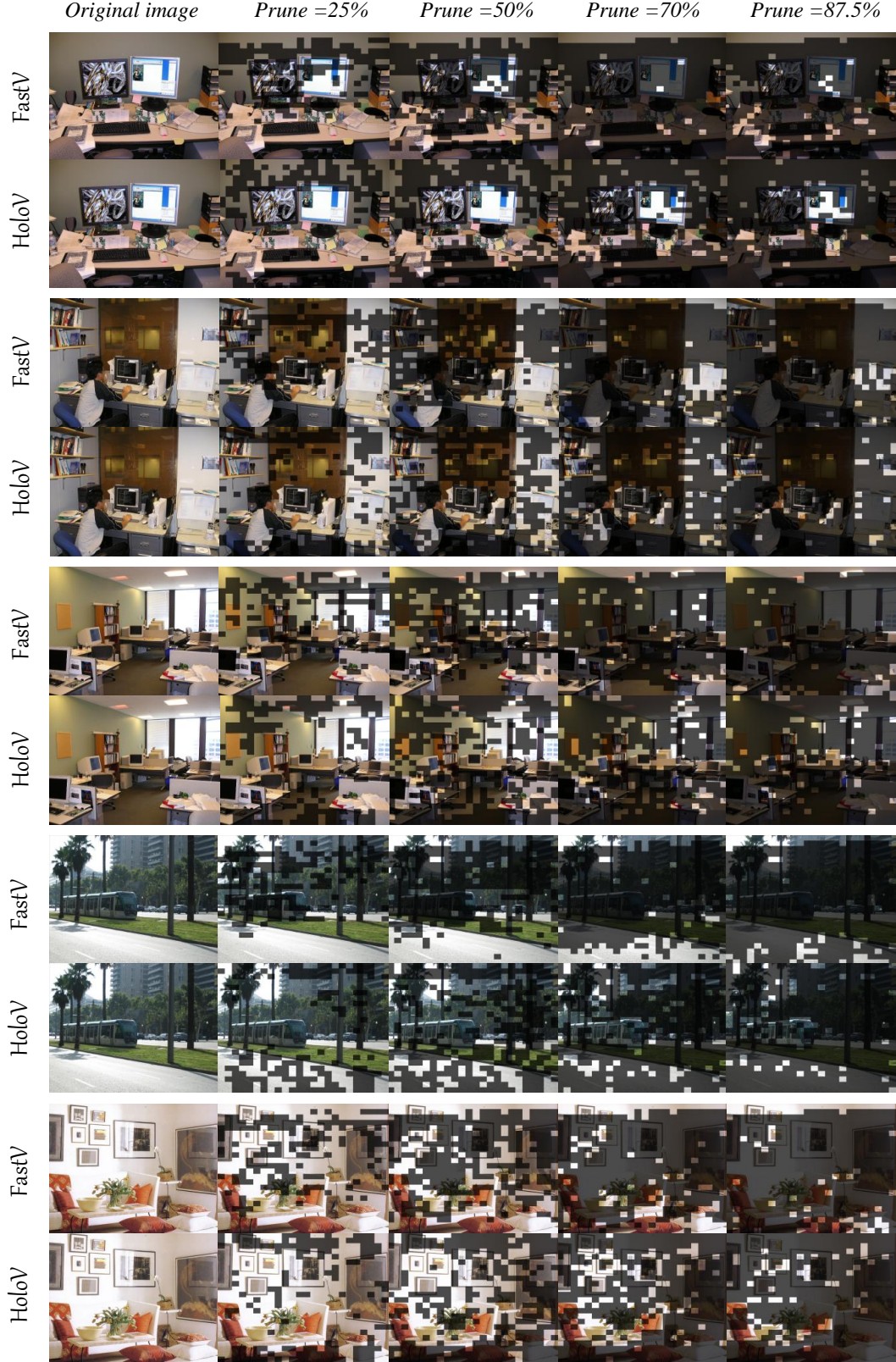

Figure 11: The case comparison between FastV and HoloV from the GQA. It presents original images alongside their pruned versions at pruning rates of 25%, 50%, 70%, and 87.5%. The bounding boxes highlight specific regions and objects across images, where HoloV well preserves the pivotal tokens.

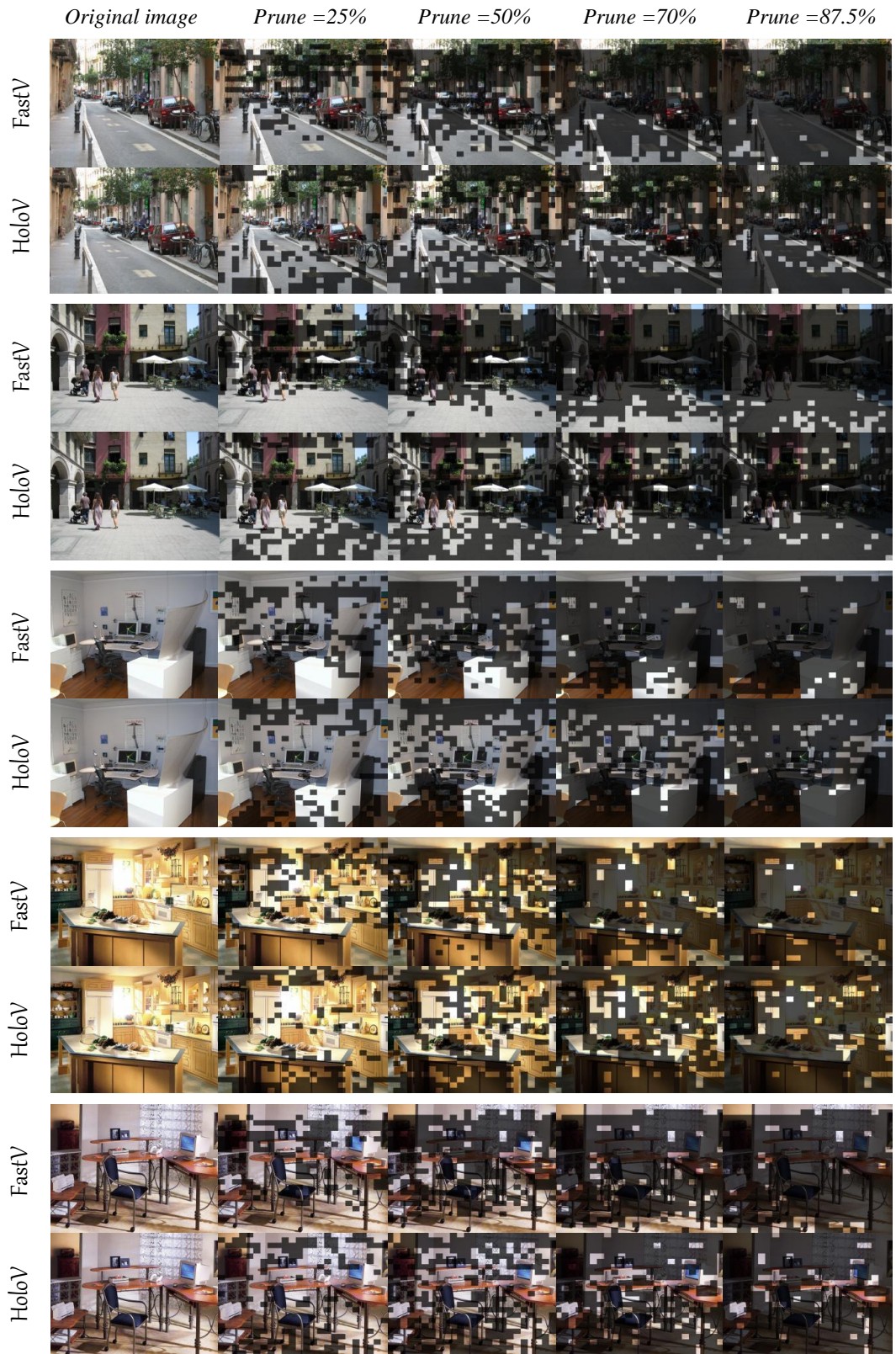

Figure 12: The case comparison between FastV and HoloV from the GQA. It presents original images alongside their pruned versions at pruning rates of 25%, 50%, 70%, and 87.5%. The bounding boxes highlight specific regions and objects across images, where HoloV well preserves the pivotal tokens.

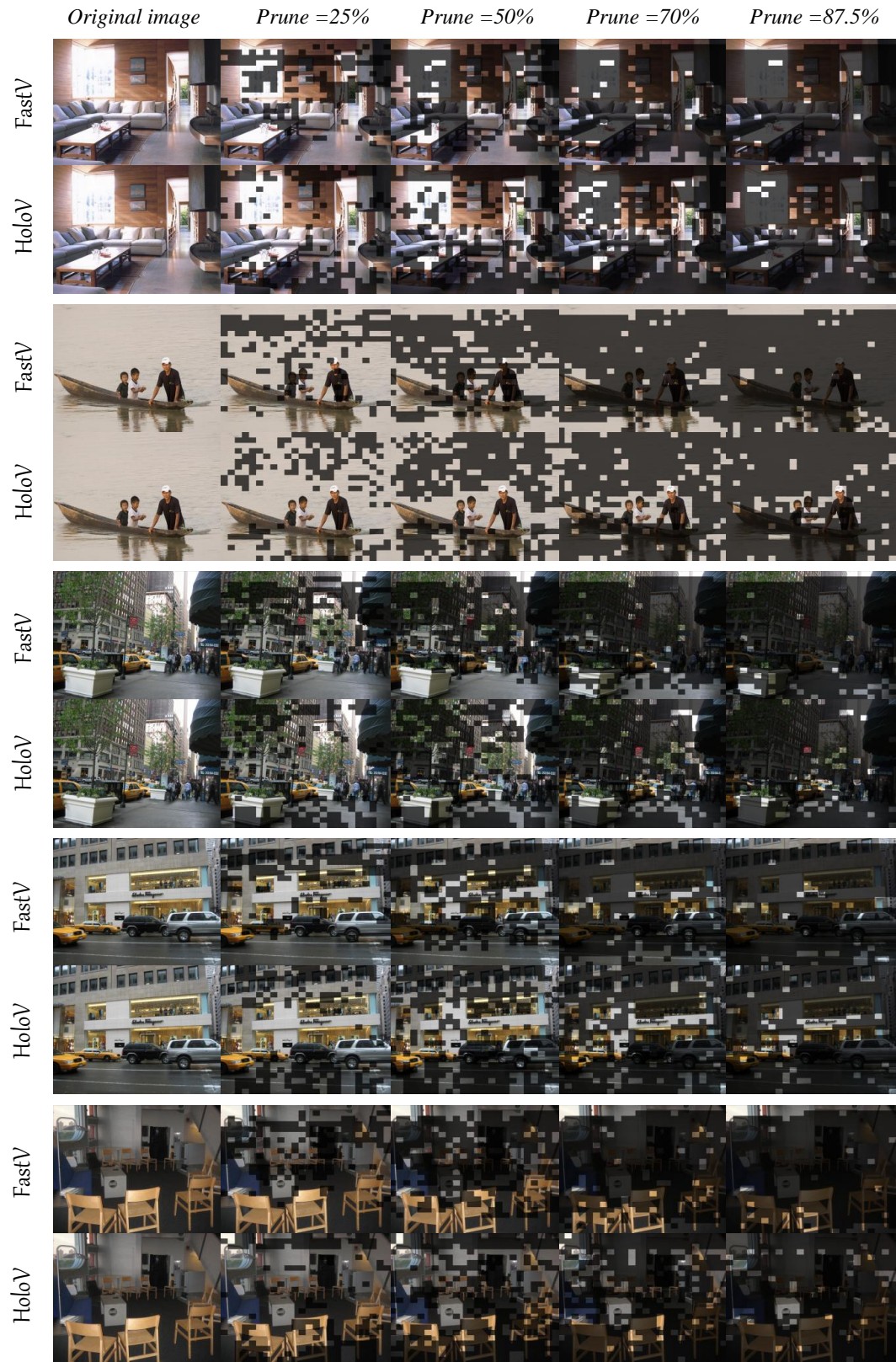

Figure 13: The case comparison between FastV and HoloV from the GQA. It presents original images alongside their pruned versions at pruning rates of 25%, 50%, 70%, and 87.5%. The bounding boxes highlight specific regions and objects across images, where HoloV well preserves the pivotal tokens.

# C  Theoretical Analysis of HoloV

To further justify the trustworthiness of our proposed HoloV, we provide a theoretical analysis of it.

**Assumption 1 (Contextual Stability)**  *Let $\mathcal{X}_v$ be the original visual tokens set, and $\mathcal{R}_v \subseteq \mathcal{X}_v$ the retained visual tokens subset, We assume the following:*

**(C1).** *For any pruned visual token $x_j \in \mathcal{X}_v \setminus \mathcal{R}_v$, there exists $x_i \in \mathcal{R}_v$ with:*

$$d(x_i, x_j) \geq \epsilon \;\; and \;\; \mathbb{V}\mathrm{ar}(d(x_i, \mathcal{N}(x_j))) \leq \delta \;,$$

*where $d$ means distance function like cosine similarity, $\mathcal{N}(x_j)$ denotes $x_j$'s local context neighbors.*

**(C2).** *For $\mathcal{H}(x_i) = \gamma \mathcal{V}(x_i) + \mathcal{A}(x_i)$ satisfies $\mathcal{H}(x_i) \geq \gamma$ for all retained tokens $x_i \in \mathcal{R}_v$*

**Lemma C.1 (Token Coverage Guarantee)**  *Under (A1), for any pruned token $x_j$, there exists $x_i \in \mathcal{R}$ such that:*

$$\|x_i - x_j\| \leq \sqrt{2(1 - \epsilon)}\|x_j\| + \sqrt{\delta}$$

**Proof C.1**  *From the cosine similarity bound, there have $x_i^\top x_j \geq \epsilon \|x_i\|\|x_j\|$. Using the variance constraint:*

$$\mathbb{E}[(x_i^\top x_k - \mu)^2] \leq \delta, \quad \forall x_k \in \mathcal{N}(x_j)$$

*where $\mu = \mathbb{E}[x_i^\top x_k]$. Combining via the triangle inequality:*

$$
\begin{aligned}
\|x_i - x_j\|^2 &= \|x_i\|^2 + \|x_j\|^2 - 2x_i^\top x_j \\
&\leq 2B^2 - 2\epsilon B^2 + \sqrt{\delta} \\
&= 2(1 - \epsilon)B^2 + \sqrt{\delta}
\end{aligned}
$$

The lemma shows that pruned tokens can be approximated by retained tokens in Euclidean space.

**Theorem C.1 (Semantic Preservation)**  *Let $f$ be a transformer layer with Lipschitz constant L. For input embeddings $\mathcal{X}_v$ and pruned set $\mathcal{R}_v$ satisfying (C1)-(C2):*

$$\|f(\mathcal{X}_v) - f(\mathcal{R}_v)\| \leq L\left[\sqrt{2(1 - \epsilon)}B + \sqrt{\delta}\right] + \eta(B, \gamma)$$

*where $\eta(B, \gamma) = \mathcal{O}\left(B^2/\gamma\right)$ is the residual error from the scoring threshold.*

**Proof C.2**  *Decompose the error into three components: 1) **Geometric distortion**: Bounded by Lemma C.1 2) **Context variance**: Controlled by $\sqrt{\delta}$ 3) **Scoring residual**:*

*For any $x \in \mathcal{X}_v \setminus \mathcal{R}_v$ with $\mathcal{S}(x) < \gamma$:*

$$\mathcal{V}^c + \mathcal{A}^c < \gamma \Rightarrow \mathcal{V}(x) < \gamma - \mathcal{A}(x)$$

*Using Cauchy-Schwarz inequality:*

$$\eta \leq \frac{1}{\gamma} \sum_{x \notin \mathcal{R}_v} \|W_V x\|^2 \leq \frac{CB^2}{\gamma}$$

*Combining terms via the triangle inequality completes the proof.*

This theorem guarantees that, even after pruning, the semantic difference between the outputs of the transformer for the original.

**Corollary 1 (Dynamic Allocation Optimality)**  *The token allocation in Section 4 achieves:*

$$\max_{\{k_p\}} \sum_{p=1}^{P} \log\left(\sum_{t=1}^{k_p} \mathcal{S}_{pt}\right) \quad s.t. \quad \sum_{p} k_p = N_{target}$$

*with approximation ratio $1 - 1/e$ when using greedy selection.*

**Proof C.3** *The allocation problem is equivalent to maximizing a monotone submodular function. Greedy algorithms provide $(1 - 1/e)$-approximation guarantees [74] for such problems.*

This corollary shows that your token allocation strategy is not only efficient but also theoretically near-optimal.

This theoretical framework demonstrates that HoloV: 1) Preserves semantic relationships through bounded geometric distortion. 2) Context variance is controlled via stability-aware pruning. 3) Token allocation is provably near-optimal, balancing efficiency and effectiveness.

# D  Fast Visual Context Refetching

## D.1  Preliminary: Reformulation of FFN

Vanilla FFN comprises two fully connected layers with non-linear activation in between. We suppose $\boldsymbol{x} \in \mathbb{R}^d$ as an input token of the FFN, and FFN function can be formulated as

$$\mathrm{FFN}(\boldsymbol{x}) = \phi\left(\boldsymbol{x}\boldsymbol{W}_1\right)\boldsymbol{W}_2^\top, \tag{7}$$

where $\phi$ is activation function like ReLU or SiLU [45], and $\boldsymbol{W}_1, \boldsymbol{W}_2 \in \mathbb{R}^{d\times D}$ are the weight matrices, in usual $D = 4d$. Peculiarly, $\boldsymbol{W}_1$ and $\boldsymbol{W}_2$ can be rewritten as

$$\boldsymbol{W}_1 = (\boldsymbol{k}_1, \boldsymbol{k}_2, \ldots, \boldsymbol{k}_D), \boldsymbol{W}_2 = (\boldsymbol{v}_1, \boldsymbol{v}_2, \ldots, \boldsymbol{v}_D), \tag{8}$$

where $\boldsymbol{k}_i, \boldsymbol{v}_i \in \mathbb{R}^d$ denote entries of key and value, respectively. As a result, the FFN can be reformulated as

$$\mathrm{FFN}(\boldsymbol{x}) = \sum \phi\left(\langle \boldsymbol{x}, \boldsymbol{k}_i \rangle\right) \cdot \boldsymbol{v}_i . \tag{9}$$

Thus, the FFN function can be construed as using input $\boldsymbol{x}$ as a query to measure similarity with keys, find matching values, and gather values by similarity, which works like a key-value memory storing the factual knowledge as found in previous studies [22, 33].

## D.2  FFN with Visual Context Refetching

We propose visual context refetching (VCR), *i.e.*, reinjecting pruned visual information into the middle layer of the text decoder during elevated uncertainty during reasoning. This strategy treats pruned visual tokens as anchors to recalibrate off-target predictions and reduces uncertainties in *object, attribute, relationship* tokens. The reason we call this pattern of reinjecting visual evidence VCR is that the model finds and refreshes key visual memories based on the hidden states in this process. In particular, inspired by the fact that FFN executes analogous retrieval from its key-value memory, we consider VCR to serve as a simplified and efficient information re-retrieval process. Given a hidden token $\boldsymbol{x} \in \mathbb{R}^d$ and dimension-aligned vision tokens $\boldsymbol{z}_v$, FFN with visual context refetching at $l$-th layer can be written as follows

$$\mathrm{FFN}^{(l)}(\boldsymbol{x} \propto \boldsymbol{z}_v) = \alpha\underline{\Delta} + (1 - \alpha)\,\mathrm{FFN}^{(l)}(\boldsymbol{x}), \tag{10}$$

where $\boldsymbol{z}_v = (\boldsymbol{z}_{v,1}, \ldots, \boldsymbol{z}_{v,N_v}) \in \mathbb{R}^{d\times N_v}$, $x \propto \boldsymbol{z}_v$ denotes execute VCR $\underline{\Delta}$ from $\boldsymbol{x}$ to visual features $\boldsymbol{z}_v$, and $\alpha \in [0, 1]$ denotes injection ratio of visual memory through the FFN layer which proportional to image complexity. Specifically, instead of performing retrieval via cross-attention layers as in previous approaches [39, 3, 102], we consider a simple retrieval process for VCR as,

$$\underline{\Delta}(\boldsymbol{z}_v \mid \boldsymbol{x}) = \sum_{i=1}^{N_v} \phi(\langle \boldsymbol{x}, \boldsymbol{z}_{v,i} \rangle) \cdot \boldsymbol{z}_{v,i}. \tag{11}$$

From the perspective of FFN, VCR works by treating $\boldsymbol{x}$ as a query, and $\langle \boldsymbol{z}_{v,i} : \boldsymbol{z}_{v,i} \rangle$ as new key-value entries (visual evidence) to supplement vision-related information in the hidden states. In this information re-retrieval process, MemVCR does not introduce any parameters that need to be trained. Notably, since the size of key-value memory $D$ in FFN typically far exceeds the number of visual tokens $N_v$ (for instance, $D = 11008$ in LLaMA-7B and $N_v = 256$ for ViT-L/14, $N_v \ll D$), the computation of VCR is negligible. Thus, VCR operation is more efficient than the cross-attention mechanism with quadratic complexity [50, 51].

### D.3  Further Efficiency Analysis

As shown in Fig. 14, we conduct efficiency evaluation on LLaVA-NeXT 7B at 95% pruning ratio, where we also introduce baseline (unpruned Vanilla) and FastV (95% pruned) for comparison. We evaluate these approaches using QA pairs from GQA, and the output length has been set to 1. During evaluation, an A800 80GB GPU has been used, and the average FLOPs, VMemory usage and throughput has been calculated, shown in Fig. 14. HoloV reduces over 90% of FLOPs requirement, 37% lower than FastV, and its VMemory usage is at the lowest level, while keeping throughput at 5.2 per second, 2.16x and 1.13x faster than baseline and FastV, respectively.

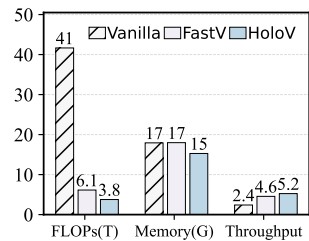

Figure 14: Inference efficiency comparison between FastV and HoloV.

## E  Impact Statement

This paper presents HoloV, a visual token pruning framework for MLLMs, and discusses its potential societal impacts. On the positive side, HoloV enhances the accessibility of multimodal technologies by reducing computational overhead, making advanced applications like medical image analysis and autonomous driving more feasible in resource-constrained environments such as edge devices or underserved regions. Its efficiency also contributes to energy sustainability by lowering the energy consumption of MLLM inference, aligning with global efforts to mitigate the environmental impact of AI. Additionally, by preserving holistic visual context instead of relying solely on attention-based "highlighted tokens," HoloV may reduce biases in model outputs, improving fairness in diverse scenarios like visual reasoning involving underrepresented communities. The framework's plug-and-play design further accelerates its integration into real-world systems, driving innovations in education, accessibility tools, and emergency response to enhance societal resilience.

However, the work also carries potential negative implications. The reduced computational barriers enabled by HoloV could facilitate misuse, such as the creation of deepfakes or misinformation, particularly in regions with limited regulatory oversight. While aiming to mitigate attention-based biases, the framework's crop-wise token allocation might inadvertently reinforce other biases if training data lacks diversity, potentially disadvantaging underrepresented groups. Moreover, the focus on inference efficiency might lead developers to prioritize speed over model interpretability, raising concerns about accountability in "black-box" deployments for high-stakes tasks like healthcare diagnostics. Lastly, over-reliance on post-hoc pruning could deter investments in more equitable training data or architectural improvements, potentially accumulating technical debt and masking foundational issues in MLLM development.

**Limitations and Future Work**.  HoloV demonstrates robust performance in preserving holistic visual context but faces two key limitations: its dependence on fixed spatial crop partitioning may hinder fine-grained semantic capture in complex scenes, and minor accuracy declines persist even at high pruning ratios (e.g., 4.2% drop when pruning 88.9% visual tokens). To address these, future work could prioritize adaptive crop, sparse attention, multi-modality extensions (*e.g.*, 3D data), and integration with hallucination mitigation, while optimizing for edge computing energy efficiency.

