# OpenReview forum: "Don't Just Chase “Highlighted Tokens” in MLLMs: Revisiting Visual Holistic Context Retention"
_NeurIPS.cc/2025/Conference — NeurIPS 2025 poster_

### Official Review · Reviewer_cqzx · 2025-07-01

**Clarity:** 3
**Significance:** 3
**Originality:** 3
**Rating:** 4
**Confidence:** 5

**Summary:**

This paper introduces HoloV, a new framework designed to improve the efficiency of multimodal large language models by intelligently pruning visual tokens. The authors observe that most existing token pruning methods focus too heavily on attention scores, often retaining only locally highlighted or similar tokens. This can lead to significant information loss and reduced performance, especially when a large proportion of visual tokens are pruned.

HoloV addresses this issue by adopting a holistic approach to token selection. Instead of simply keeping the most salient tokens based on attention, the framework divides the image into spatial crops and adaptively selects tokens from different regions. This ensures that both global context and diverse local details are preserved, even when aggressive pruning is applied. The method balances local saliency with overall scene understanding, aiming to maintain important semantic information that would otherwise be lost.

**Questions:**

1. Can you provide detailed information on how spatial crops, crop numbers, and related hyperparameters are selected? Additionally, could you include ablation studies that show how these choices impact the reproducibility and performance of your approach?

2. What is the rationale behind your specific top-k token selection score? How does this formulation compare empirically or theoretically to alternative selection strategies or scoring methods?

3. Have you evaluated the sensitivity and robustness of your adaptive holistic token allocation procedure, including the effects of discretization and the sharpness parameter? Could you discuss or provide results comparing your allocation approach to other possible methods?

4. Would it be possible to use HoloV as a replacement for the Q-Former module in models such as InstructBLIP, or to integrate HoloV within the Q-Former family of architectures? If so, could you discuss any practical considerations, challenges, or potential benefits of applying HoloV in this context? It would be helpful to understand whether HoloV is compatible with Q-Former-based models and how its holistic token pruning strategy might interact with or improve these architectures.

5. Could you report practical efficiency metrics such as multiply-accumulate operations (MACs), wall-clock latency, and memory usage on different hardware platforms, including less powerful or older devices and different devices (GPUs with different floating point operations)? It would be helpful to see evidence of real-world efficiency beyond theoretical FLOPs reductions.

6. Can you present empirical results or ablation studies that isolate and quantify the actual contribution of visual context refetching (VCR) in your framework? In particular, how is the injection ratio chosen, and what impact does VCR have on uncertainty reduction or downstream performance?

7. Do you have results or ablation studies demonstrating how HoloV performs with a wider variety of visual encoders, such as SigLIP, DINOv2, CLIP-ResNet,S2, Mamba, ConvNeXt, etc? Would you consider evaluating whether your method generalizes to models beyond the LLaVA family?

**Ethical Concerns:**

["NO or VERY MINOR ethics concerns only"]

**Final Justification:**

Given what was discussed during the rebuttal, some of the questions were resolved. I encourage the authors to edit the final document by incorporating all the suggestions presented, with special emphasis on the analysis of compute overhead.

**Limitations:**

See questions and weakness.

**Paper Formatting Concerns:**

Several tables and figures in the paper do not follow the official NeurIPS template and formatting guidelines. In particular, Figure 5, Figure 6, Figure 9, Table 3, and Table 4 stand out as not being consistent with the expected style. I recommend revising these figures and tables to conform with the NeurIPS template.

**Quality:**

3

**Strengths And Weaknesses:**

### Strenghts
* The paper presents comprehensive experiments across a wide range of benchmarks and MLLM architectures, demonstrating consistent and significant improvements over state-of-the-art token pruning methods.
* The paper provides a well-articulated motivation for holistic context retention and offers insightful analysis of the limitations of prior methods, supported by visualizations and ablation studies.
* HoloV is easy to integrate, offering substantial computational efficiency gains with minimal performance loss.

### Weaknesses
* Key implementation details, such as: how spatial crops are defined, the choice of hyperparameters, and ablation on the crop number or allocation sharpness, are insufficiently specified. This limits reproducibility and undermines the credibility of the claimed results.
* The process for selecting the top-k visual tokens within each crop is largely based on a straightforward score maximization, but the underlying criteria for combining local saliency and global relevance remain rather heuristic and are not thoroughly justified or compared to alternative strategies. There is no empirical evidence or ablation provided to show how sensitive the method is to the exact formulation of the selection score, or whether more sophisticated selection strategies could lead to further improvements.
* The adaptive holistic token allocation approach in HoloV is largely heuristic and lacks principled justification. The process involves averaging token scores within each crop, discretizing quotas, and then using iterative reallocation to handle rounding issues or quota overflow. While this ensures that each region of the image is represented, the method relies on several arbitrary choices, such as the setting of the sharpness parameter $\tau$ and the specific reallocation rules. The paper does not provide any analysis or ablation to show how sensitive the method is to these choices, nor does it offer a theoretical rationale for why this allocation procedure is optimal. As a result, the effectiveness and robustness of this adaptive allocation are uncertain, and it remains unclear whether the improvements are due to this specific mechanism or would be matched by simpler or more principled alternatives.
* While the paper provides a theoretical analysis of computational complexity using FLOPs, it does not sufficiently address how pruning affects actual performance on real hardware. In practice, efficiency gains from pruning are highly dependent on multiply-accumulate operations (MACs), memory access patterns, and especially system latency, which can vary significantly depending on the hardware architecture. On older or resource-constrained devices, aggressive pruning may actually introduce irregular computation and reduce parallelism, potentially resulting in longer inference times or even hardware underutilization. The paper does not empirically explore how HoloV’s pruning strategy impacts different hardware setups, nor does it report detailed results on wall-clock latency or MACs. For a complete and convincing assessment of practical efficiency, future work should explicitly measure and report latency and MACs across diverse hardware platforms, since improvements in FLOPs alone do not guarantee real-world speedup or energy savings.
* I find the section on **visual context refetching (VCR)** to be interesting in theory, but I am concerned about the lack of concrete empirical evidence supporting its utility. While the mathematical description of how pruned visual tokens are reintegrated via the feed-forward network is clearly presented, the paper does not provide experiments or detailed analysis to demonstrate the real impact of this mechanism. Specifically, the authors do not explain how the injection ratio is selected in practice, and there is no data illustrating how VCR affects uncertainty reduction or downstream performance. Without a dedicated evaluation or ablation that isolates the effect of VCR, it is challenging to judge how important or reliable this component truly is within the broader HoloV framework.
* While the paper provides results for both LLaVA-1.5 and LLaVA-NeXT visual encoders, the experimental analysis remains limited to this family of encoders. There is no investigation into how HoloV would perform with fundamentally different visual backbones, such as other types of vision transformers (Siglip2, Dinov2, etc..), convolutional encoders (CLIP-Resnet,ConvNeXt, EfficientNet, etc..), or others models (Mamba). Since different encoders may have unique tokenization strategies and varying levels of feature redundancy, it is unclear if HoloV’s advantages and efficiency–accuracy trade-offs would generalize beyond the LLaVA architecture. A broader ablation study across diverse encoder types would strengthen the paper and provide more confidence in the general applicability of the method.

---

> ### Author Rebuttal · Authors · 2025-07-31
>
> We sincerely thank Reviewer cqzx for the insightful comments.
>
> ---
>
> > Q1: Can you provide detailed information on how spatial crops, crop numbers, and related hyperparameters are selected? Additionally, could you include ablation studies that show how these choices impact the reproducibility and performance of your approach?
>
> We show performance when we vary the total number of crops from 4 to 24, and we observe that varying the number of crops in the range of 4-16 does not have a significant effect on performance, but if the number of patches is too large (>20), it causes a decrease in performance.
>
> |Patch_num	|4	|8	|12	|16	|20	|24|
> | :---: | :---: | :---: | :---: | :---: | :---: | :---: |
> |Retain 64 tokens	|89.3%	|89.3%|	90.0%|	91.2%|	91.3%|	90.2%|
> |Retain 128 tokens	|94.5%	|95.1%|	94.6%|	94.8%|	95.6%|	95.9%|
> |Retain 192 tokens	|95.1%	|96.7%|	96.1%|	94.9%|	92.4%|	91.3%|
>
> *where the numbers indicate the percentage of performance relative to the raw performance.*
>
> For hyperparameters, we usually set allocation sharpness $\tau=1$, and set the same value in all experiments.
>
> ---
>
> > Q2: What is the rationale behind your specific top-k token selection score? How does this formulation compare empirically or theoretically to alternative selection strategies or scoring methods?
>
> Our objective is $\mathcal{H}^c =  \gamma_c \mathcal{V}^c + \mathcal{A}^c$. Following your suggestion, we divided the objective into local saliency $\mathcal{V}$ and global relevance $\mathcal{A}$, and we compare against pure local saliency, global relevance, and our HoloV. The results when retaining 64 tokens are as follows,
>
> | Strategy | MME | POPE | SQA | MMB-EN|
> | :---: |  :---: |  :---: |  :---: |  :---: |
> |local saliency|1604|68.9|65.3|60.1|
> |global relevance|1687|75.8|68.9|60.6|
> |HoloV|1715|80.3|69.5|63.3|
>
> The results demonstrate that HoloV's strategy is reasonable. About whether more sophisticated selection strategies could lead to further improvements, we are researching a text-based approach to further improve the performance of HoloV.
>
> ---
>
> > Q3: Have you evaluated the sensitivity and robustness of your adaptive holistic token allocation procedure, including the effects of discretization and the sharpness parameter? Could you discuss or provide results comparing your allocation approach to other possible methods?
>
> While it needs setting of the sharpness parameter, we set $\tau=1$ in all experiments, and we also tried setting other values, and the results show that the model is not sensitive to this hyperparameter. For the effectiveness and robustness of this adaptive allocation, we vary the total number of crops from 4 to 16, and we observe that varying the number of crops does not have a significant effect on performance.
>
> |Patch_num	|4	|8	|12	|16	|
> | :---: | :---: | :---: | :---: | :---: |
> |Retain 64 tokens	|89.3%	|89.3%|	90.0%|	91.2%|
> |Retain 128 tokens	|94.5%	|95.1%|	94.6%|	94.8%|
> |Retain 192 tokens	|95.1%	|96.7%|	96.1%|	94.9%|
>
> ---
>
> > Q4: Would it be possible to use HoloV as a replacement for the Q-Former module in models such as InstructBLIP, or to integrate HoloV within the Q-Former family of architectures? If so, could you discuss any practical considerations, challenges, or potential benefits of applying HoloV in this context? It would be helpful to understand whether HoloV is compatible with Q-Former-based models and how its holistic token pruning strategy might interact with or improve these architectures.
>
> To integrate HoloV within the Q-Former family of architectures, we conducted experiments on Qwen2.5-VL.
>
> |Method	|MME|	POPE|	MMB-EN|	SQA|	Average|
> | :-----: |  :-----: |  :-----: |  :-----: |  :-----: |  :-----: |
> |Qwen2.5-VL-7B	|2304	|86.1	|82.8	|84.7	|100%|
> |FastV (prune 66.7%)	|2072	|82.2	|75.7	|78.5	|92.37%|
> |HoloV (prune 66.7%)	|2093	|85	|78.3	|79.8	|94.58%|
> |FastV (prune 77.8%)	|2036	|80.7	|74.9	|78	|91.16%|
> |HoloV (prune 77.8%)	|2043	|82.3	|76.5	|79.8	|92.71%|
> |FastV (prune 88.9%)	|1940	|78.6	|69.2	|77.4	|87.96%|
> |HoloV (prune 88.9%)	|2006	|80.7	|72.4	|79.5	|90.52%|
>
> These results demonstrate that HoloV achieves superior performance.
>
> ---
>
> > Q5: Could you report practical efficiency metrics such as multiply-accumulate operations (MACs), wall-clock latency, and memory usage on different hardware platforms, including less powerful or older devices and different devices (GPUs with different floating point operations)? It would be helpful to see evidence of real-world efficiency beyond theoretical FLOPs reductions.
>
> We reported  practical efficiency metrics on 4060Ti GPU (16GB), the results are as follows,
>
> |Method| Time| Prefill|Latency(s)|Memory| Average Peformance.|
> | :---:| :---:| :---:| :---:| :---:| :---:|
> |*Vanilla (576 tokens)*|49:41|0.5ms|0.334|19.0GB|100%|
> |FasterVLM (192 tokens)|30:09|0.6ms|0.202|15.6GB|98.5%|
> |FastV (192 tokens)|35:34|0.5ms|0.239|16.0GB|90.5%|
> |HiRED (192 tokens)|30:08|0.6ms|0.202|15.6GB|94.6%|
> |MustDrop (192 tokens)|32:30|0.5ms|0.218|15.6GB|97.2%|
> |SparseVLM  (192 tokens)|37:20|0.5ms|0.251|15.6GB|96.1%|
> |HoloV  (192 tokens)|31:02|0.6ms|0.208|15.6GB|99.2%|
> |FasterVLM (58 tokens)|25:08|0.6ms|0.168|14.5GB|92.5%|
> |FastV (58 tokens)|30:41|0.5ms|0.206|15.6GB|76.7%|
> |HiRED (58 tokens)|25:03|0.6ms|0.168|14.5GB|89.4%|
> |MustDrop (58 tokens)|29:40|0.5ms|0.199|14.5GB|90.1%|
> |SparseVLM  (58 tokens)|31:28|0.5ms|0.212|14.6GB|87.3%|
> |HoloV  (58 tokens)|28:28|0.5ms|0.176|14.5GB|95.7%|
>
> ---
>
> > Q6: Can you present empirical results or ablation studies that isolate and quantify the actual contribution of visual context refetching (VCR) in your framework? In particular, how is the injection ratio chosen, and what impact does VCR have on uncertainty reduction or downstream performance?
>
> Sure. We added experiments about the isolated contribution of visual context refetching (VCR) in our framework and under different injection ratios.
>
> | Strategy | MME | POPE | SQA | MMB-EN|
> | :---: |  :---: |  :---: |  :---: |  :---: |
> |HoloV w/o VCR|1700|78.3|69.2|62.0|
> |HoloV|1715|80.3|69.5|63.3|
>
> | Strategy | MME | POPE | SQA | MMB-EN|
> | :---: |  :---: |  :---: |  :---: |  :---: |
> |HoloV (0.05) |1702|78.5|69.2|62.3|
> |HoloV (0.10)|1708|79.9|69.3|62.8|
> |HoloV (0.15)|1715|80.3|69.5|63.3|
> |HoloV (0.20)|1711|80.3|69.4|63.0|
> |HoloV (0.25)|1708|80.1|69.5|63.2|
> |HoloV (0.30)|1705|79.7|69.3|62.9|
>
> For the injection ratio, we often set it to 0.15, and this strategy can effectively contribute to the performance improvements.
>
> ---
>
> > Q7: Do you have results or ablation studies demonstrating how HoloV performs with a wider variety of visual encoders, such as SigLIP, DINOv2, CLIP-ResNet,S2, Mamba, ConvNeXt, etc? Would you consider evaluating whether your method generalizes to models beyond the LLaVA family?
>
> We have not conducted ablation studies with a wider variety of visual encoders as time is too short and this needs pretraining these visual encoders (most MLLMs' visual encoders are ViT framework), in the future we will take time to research such influence. Following your suggestion, we have additionally included experiments with Qwen2.5-VL (beyond the LLaVA family), and the results on multiple benchmarks demonstrate HoloV's effectiveness. (*Table in response to Q4*)
>
> ---
>
> > Q8: Paper Formatting Concerns: Several tables and figures in the paper do not follow the official NeurIPS template and formatting guidelines. In particular, Figure 5, Figure 6, Figure 9, Table 3, and Table 4 stand out as not being consistent with the expected style. I recommend revising these figures and tables to conform with the NeurIPS template.
>
> We have reformatted **all figures and tables** to match the NeurIPS style guidelines—single-column width for tables, consistent font sizes, correct caption placement, and vector‐format graphics.

---

> > ### Comment · Reviewer_cqzx · 2025-08-06
> > **Response to the authors: PART 1**
> >
> > **Q1**: The question was resolved.
> >
> > **Q2**: The question was resolved.
> >
> > **Q3**: The question was resolved.
> >
> > **Q4**:
> >
> > While the authors describe integrating HoloV into the Q-Former family and conducting experiments on Qwen2.5-VL, it's important to note that **Qwen2.5-VL does not include a Q-Former module**. I encourage the authors to revisit the architecture details presented in [this paper](https://arxiv.org/abs/2502.13923) to verify this point.
> >
> > More importantly, I’m interested in understanding how HoloV compares to **Q-Former-based methods**, which explicitly learn a set of query embeddings to **compress visual features via attention**. At the end of the day, a Q-Former learns parameters that serve as a mechanism for visual abstraction **(remove visual tokens)** .
> >
> > I truly believe that emphasizing this comparison would **strengthen the exposition of HoloV’s contributions**, particularly in relation to the visual token compression paradigm that Q-Former architectures exemplify.

---

> ### Comment · Reviewer_cqzx · 2025-08-06
> **Response to the authors: PART 2**
>
> **Q5**:
> In computing, the execution time of a piece of code depends on many factors: hardware factors like the CPU and GPU, and software factors such as OS versions and Python dependencies, among others. Therefore, quantitatively, having execution time alone is not a universal metric to consider. **That’s why I asked about FLOPS and MACs using Holo and the baseline without Holo**. You don’t need to run multiple comparisons with various token lengths; you can use a smaller number of tokens, which is the fastest/efficiency can be backed up with numbers.
>
> **For latency, you should run one example only and use profiling like this:**
>
> ```
> start = torch.cuda.Event(enable_timing=True)
> end   = torch.cuda.Event(enable_timing=True)
>
> start.record()
> output = model(input)
> end.record()
>
> # Wait for all kernels to finish
> torch.cuda.synchronize()
> print(start.elapsed_time(end), "ms")
> ```
> **For MACS and FLOPS:**
> | Package                      | Link                                                                                                                             |
> | ---------------------------- | -------------------------------------------------------------------------------------------------------------------------------- |
> | **THOP**                     | [ultralytics/thop](https://github.com/ultralytics/thop) ([GitHub][1])                                                            |
> | **ptflops**                  | [LukasHedegaard/ptflops](https://github.com/LukasHedegaard/ptflops) ([GitHub][2])                                                |
> | **PyTorch-OpCounter**        | [Lyken17/pytorch-OpCounter](https://github.com/Lyken17/pytorch-OpCounter) ([GitHub][3])                                          |
> | **flops-counter.pytorch**    | [sovrasov/flops-counter.pytorch](https://github.com/sovrasov/flops-counter.pytorch) ([GitHub][4])                                |
> | **torchprofile**             | [zhijian-liu/torchprofile](https://github.com/zhijian-liu/torchprofile) ([GitHub][5])                                            |
> | **fvcore FlopCountAnalysis** | [facebookresearch/fvcore ﬂop\_count.md](https://github.com/facebookresearch/fvcore/blob/master/docs/flop_count.md) ([GitHub][6]) |
> | **calculate-flops.pytorch**  | [MrYxJ/calculate-flops.pytorch](https://github.com/MrYxJ/calculate-flops.pytorch) ([GitHub][7])                                  |
> | **torch-flops**              | [torch-flops on PyPI](https://pypi.org/project/torch-flops/) ([PyPI][8])                                                         |
> | **torchstat**                | [torchstat on PyPI](https://pypi.org/project/torchstat/) ([PyPI][9])                                                             |
> | **flops-profiler**           | [cli99/flops-profiler](https://github.com/cli99/flops-profiler) ([GitHub][10])                                                   |
>
> [1]: https://github.com/ultralytics/thop?utm_source=chatgpt.com "ultralytics/thop: Profile PyTorch models for FLOPs and ..."
> [2]: https://github.com/LukasHedegaard/ptflops?utm_source=chatgpt.com "LukasHedegaard/ptflops: Flops counter for convolutional ..."
> [3]: https://github.com/Lyken17/pytorch-OpCounter?utm_source=chatgpt.com "Lyken17/pytorch-OpCounter: Count the MACs / FLOPs ..."
> [4]: https://github.com/sovrasov/flops-counter.pytorch?utm_source=chatgpt.com "Flops counter for neural networks in pytorch framework"
> [5]: https://github.com/zhijian-liu/torchprofile?utm_source=chatgpt.com "zhijian-liu/torchprofile: A general and accurate MACs / ..."
> [6]: https://github.com/facebookresearch/fvcore/blob/master/docs/flop_count.md?utm_source=chatgpt.com "Flop Counter for PyTorch Models - facebookresearch/fvcore"
> [7]: https://github.com/MrYxJ/calculate-flops.pytorch?utm_source=chatgpt.com "calflops: a FLOPs and Params calculate tool for neural ..."
> [8]: https://pypi.org/project/torch-flops/?utm_source=chatgpt.com "torch-flops"
> [9]: https://pypi.org/project/torchstat/?utm_source=chatgpt.com "torchstat"
> [10]: https://github.com/cli99/flops-profiler?utm_source=chatgpt.com "cli99/flops-profiler: pytorch-profiler"

---

> > ### Comment · Reviewer_cqzx · 2025-08-06
> > **Response to the authors: PART 3**
> >
> > **Q6:** What’s your intuition for why **0.15** yields the best performance? Was it something empirical?
> >
> > **Q7:**  The question was resolved.
> >
> > **Q8:** The question was resolved.
> >
> > **The paper has a lot of potential; I hope you can address the remaining questions, and I’ll happily raise the rating.**

---

### Official Review · Reviewer_h9EW · 2025-07-02

**Clarity:** 3
**Significance:** 2
**Originality:** 3
**Rating:** 5
**Confidence:** 2

**Summary:**

This paper presents HoloV, a novel visual token pruning framework for MLLMs, which addresses the critical limitation of attention-based methods that prioritize local saliency over global semantic context. By introducing crop-wise adaptive allocation and a hybrid scoring mechanism combining attention saliency and semantic variance, HoloV ensures holistic context retention during aggressive pruning.

**Questions:**

Please see the Major points listed under Weaknesses.
How does the granularity of crop partitioning (e.g., number of crops) affect pruning efficiency? For high-resolution images (e.g., LLaVA-NeXT's 2,880 tokens), should the number of crops be dynamically adjusted?

**Ethical Concerns:**

["NO or VERY MINOR ethics concerns only"]

**Final Justification:**

The authors has addressed most of my concerns. While some minor reservations remain, I believe the paper's contribution is significant and meets the bar for acceptance. Therefore, I have raised my score to Accept.

**Limitations:**

When an LLM has severe language bias, could HoloV's visual pruning indirectly exacerbate hallucinations? For example, if the language model misjudges an object's existence, might HoloV retain irrelevant visual tokens that reinforce this error?

**Quality:**

3

**Strengths And Weaknesses:**

**Strengths:**
1. This paper provides a solid motivation for addressing visual token pruning in MLLMs. Traditional attention-based approaches tend to fragment global semantics by overly focusing on locally emphasized tokens.  By incorporating insights from cognitive science and analyzing Positional Bias, the paper highlights the root causes of performance degradation in complex scenes.
2. HoloV shows exceptional control over performance degradation, retaining 95.8% of the original performance after pruning 88.9% of visual tokens.
3.The paper is well-organized and easy to follow.

**Weaknesses:**

1. The fixed spatial crop partitioning method limits the system's applicability in complex scenes where semantic regions may be non-rectangular or cross boundaries. This can result in fragmented semantic units, reducing the model’s generalizability.
2. There is also a sensitivity to hyperparameters like the sharpness factor (τ) and scaling factor (γ), which require manual tuning. The lack of a unified optimization strategy for varying resolutions or scene complexities increases deployment costs and limits real-world applicability.
3. While HoloV mitigates Positional Bias, it still struggles in extreme conditions such as severe lighting, occlusions, or fast-moving objects. These challenges lead to potential information loss, as seen in MSRVTT-QA experiments where pruning errors affected dynamic object understanding.

---

> ### Author Rebuttal · Authors · 2025-07-31
>
> We sincerely thank Reviewer h9EW for the constructive comments on our work. Below, we will address each of your questions and concerns.
>
> ---
>
> > Q1: The fixed spatial crop partitioning method limits the system's applicability in complex scenes where semantic regions may be non-rectangular or cross boundaries. This can result in fragmented semantic units, reducing the model’s generalizability.
>
> Actually, the quota of a crop is dynamically allocated, thus, this would not affect flexible allocation based on the density or complexity of the content in the image, which could reduce fragmentation and improve model performance in more intricate scenes.
>
> ---
>
> > Q2: There is also a sensitivity to hyperparameters like the sharpness factor (τ) and scaling factor (γ), which require manual tuning. The lack of a unified optimization strategy for varying resolutions or scene complexities increases deployment costs and limits real-world applicability.
>
> We acknowledge that *τ* (sharpness factor) requires manual setting, but in practice, these hyperparameters do not have a significant effect on the results. For example, the sharpness factor (τ) is always set to 1. For $\gamma_c$ (scaling factor), it is determined by $\mathbb{E}[\|\mathcal{A}^c\|]/\mathbb{E}[\|\mathcal{V}^c\|]$.
>
> ---
>
> > Q3: While HoloV mitigates Positional Bias, it still struggles in extreme conditions such as severe lighting, occlusions, or fast-moving objects. These challenges lead to potential information loss, as seen in MSRVTT-QA experiments where pruning errors affected dynamic object understanding.
>
> Thank you for your recognition of our work. We agree that extreme conditions like severe lighting, occlusions, and fast-moving objects pose significant challenges for HoloV and can lead to information loss. In response, we are considering temporal pruning, i.e., tracking **dynamic objects** across frames, pruning **static duplicates**, to better capture dynamic changes and reduce pruning errors in such conditions. This strategy would improve the robustness of HoloV in challenging scenarios and dynamic object understanding, where traditional image models may struggle to capture the full range of information.
>
> ---
>
> > Q4: How does the granularity of crop partitioning (e.g., number of crops) affect pruning efficiency? For high-resolution images (e.g., LLaVA-NeXT's 2,880 tokens), should the number of crops be dynamically adjusted?
>
> In fact, the granularity of pruning partitions does not affect pruning efficiency, as the retained visual markers are determined by pruning quotas, and the quota in a crop is calculated by the informativeness of intra-crop visual tokens, which is dynamic, the total pruning quotas would not be changed. For high-resolution images, we agree that adjusting the number of crops dynamically could be beneficial. For instance, using a lower number of crops for highly detailed areas (such as objects or regions of interest) and a higher number of crops for less detailed regions (such as background areas) could improve pruning performance.
>
> Further, we show performance when we vary the total number of crops from 4 to 16, and we observe that varying the number of crops does not have a significant effect on performance.
>
> |Patch_num	|4	|8	|12	|16|
> | :---: |  :-----: |  :-----: |  :-----: |  :-----: |
> |*Vanilla (576 tokens)*	|*100%*	|*100%*	|*100%*	|*100%*	|
> |Retain 64 tokens	|89.3%	|89.3%	|90.0%	|91.2%|
> |Retain 128 tokens	|94.5%	|95.1%	|94.6%	|94.8%|
> |Retain 192 tokens	|95.1%	|96.7%	|96.1%	|94.9%|
>
> *where the numbers indicate the percentage of performance relative to the raw performance.*
>
> ---
>
> > Q5: When an LLM has severe language bias, could HoloV's visual pruning indirectly exacerbate hallucinations? For example, if the language model misjudges an object's existence, might HoloV retain irrelevant visual tokens that reinforce this error?
>
> We recognize that when an LLM has severe language bias (e.g., in tasks involving object recognition or reasoning), it may hallucinate or misjudge the existence of certain objects. If this occurs, visual pruning could indeed exacerbate the issue by retaining irrelevant visual tokens. However, in the experiments, especially in POPE, which is a hallucination-related benchmark, compared with SOTA methods, our HoloV achieves the best performance, as shown in the table.
>
> |Method	|Retain 192 tokens	|Retain 128 tokens	|Retain 64 tokens|
> | :---: | :---: | :---: | :---: |
> |Vanilla (576 tokens)	|85.9	|85.9	|85.9|
> |ToMe	|72.4	|62.8	|52.5|
> |FastV	|64.8	|59.6	|48.0|
> |LLaVA-PruMerge	|71.3	|67.2	|65.3|
> |PDrop	|82.3	|82.3	|55.9|
> |FiCoCo-V	|82.5	|82.2	|76.0|
> |MustDrop	|82.6	|78.7	|68.0|
> |HiRED	|82.8	|79.8	|73.6|
> |SparseVLM	|83.6	|80.5	|75.1|
> |HoloV (ours)	|85.6	|84.0	|80.3|
>
> The results demonstrate that HoloV's visual pruning has minimal impact on exacerbating hallucinations.
>
> In the future, we aim to explore cross-modal consistency checks, where visual tokens are only retained if they align with the language model's predictions. If the LLM's prediction is highly uncertain, we would retain more visual tokens associated with that object, potentially reducing hallucinations.
>
> We appreciate the reviewer’s thoughtful feedback on these critical aspects of HoloV. We look forward to continuing to refine HoloV for real-world applicability.

---

> > ### Comment · Reviewer_h9EW · 2025-08-06
> > **General  response to the authors**
> >
> > Thank you for your detailed rebuttal. The clarification on hyperparameters and the supplementary data regarding crop granularity and potential hallucinations are convincing and have addressed my concerns.
> >
> > While the discussion on fixed spatial partitioning and performance under extreme conditions could still be improved, I want to clarify my core concern for Q1. My reservation was spatial in nature: the visual information of an object (for example, a person tilted at an angle) could be physically partitioned across two or more grid regions by the grid lines. If one of these regions is then deemed unimportant (e.g., containing only the person's feet and some background), this information would be discarded, leading to incomplete object semantics.
> >
> > Nevertheless, considering the strong evidence provided elsewhere in your response, I recognize the paper's overall contribution and am willing to raise my score. I recommend that you integrate the new data and analysis provided for Q4 and Q5 into the final version of the paper or its appendix.

---

> > > ### Author Response · Authors · 2025-08-07
> > >
> > > Dear Reviewer h9EW,
> > >
> > > Thank you for your recognition of our work. We sincerely appreciate your feedback. Your constructive comments have been instrumental in enhancing our work. We have integrated the new data and analysis provided for Q4 and Q5 into the final version of the paper. And we will further improve our work, especially in terms of fixed spatial partitioning and performance under extreme conditions.
> > >
> > > Thank you again for your time and thoughtful consideration.
> > >
> > > Best wishes,
> > >
> > > Authors

---

### Official Review · Reviewer_4HaK · 2025-07-02

**Clarity:** 3
**Significance:** 3
**Originality:** 3
**Rating:** 5
**Confidence:** 3

**Summary:**

This paper introduces HoloV, a plug-and-play visual token pruning framework for MLLMs that moves beyond attention-based token selection by preserving holistic visual context. Unlike previous methods that focus on high-attention tokens, HoloV adaptively allocates pruning budgets across spatial crops, balancing local saliency with global semantic diversity. The method significantly reduces visual tokens (up to 88.9%) while retaining up to 96% of the original performance, improving inference efficiency across multiple benchmarks and models.

**Questions:**

### Question:
1. I would like to know whether HoloV can still achieve consistent strong performance on models using native resolution (e.g., Qwen2.5-VL). Adding such results would allow HoloV to cover all mainstream VLM visual encoding forms, further enhancing the persuasiveness of the results.

**Ethical Concerns:**

["NO or VERY MINOR ethics concerns only"]

**Final Justification:**

My question has been answered, and the author has provided more comprehensive results to respond. I'll keep my score to support their work.

**Limitations:**

yes

**Quality:**

3

**Strengths And Weaknesses:**

### Strength:
1. The paper is detailed and clear, with thorough experiments, visualizations, and theoretical analysis that make the claims convincing.
2. Experiments on rich benchmarks with various retained token numbers show promising improvements, demonstrating the effectiveness of the proposed method.

### Weakness:
1. The evenly partitioned crops may not be flexible enough and could introduce potential spatial bias.

---

> ### Author Rebuttal · Authors · 2025-07-31
>
> We sincerely thank Reviewer 4HaK for the constructive comments on our work. We are very grateful to the reviewer for recognising the novelty of our idea and the richness and rationality of our experiments.
>
> ---
>
> > Q1: The evenly partitioned crops may not be flexible enough and could introduce potential spatial bias.
>
> Thanks. Although crops are partitioned evenly, the quota of a crop is dynamically allocated, thus, this would not affect flexible allocation based on the density or complexity of the content in the image, which could reduce fragmentation and improve model performance in more intricate scenes.
>
> To further show the advantages of our method, we further present our results on MM-Vet benchmark as follows,
>
> | Method | Retain 192 tokens | Retain 128 tokens |	Retain 64 tokens |
> | :-----: |  :-----: |  :-----: |  :-----: |
> |*Vanilla (576 tokens)*	|*31.0*|*31.0*	|*31.0*|
> |FastV	|28.9	|26.7	|26.1|
> |SparseVLM	|31.5	|30.0	|23.3|
> |VisionZip	|31.7	|32.6	|30.2|
> |HoloV (ours)	|32.9	|32.0	|30.2|
>
> ---
>
> > Q2: I would like to know whether HoloV can still achieve consistent strong performance on models using native resolution (e.g., Qwen2.5-VL). Adding such results would allow HoloV to cover all mainstream VLM visual encoding forms, further enhancing the persuasiveness of the results.
>
> We agree with the importance of having HoloV cover all major VLMs, and based on your suggestion we have additionally included experiments with Qwen2.5-VL, and the results on multiple benchmarks demonstrate HoloV's effectiveness.
>
> |Method	|MME|	POPE|	MMB-EN|	SQA|	Average|
> | :-----: |  :-----: |  :-----: |  :-----: |  :-----: |  :-----: |
> |Qwen2.5-VL-7B	|2304	|86.1	|82.8	|84.7	|100%|
> |FastV (prune 66.7%)	|2072	|82.2	|75.7	|78.5	|92.37%|
> |HoloV (prune 66.7%)	|2093	|85	|78.3	|79.8	|94.58%|
> |FastV (prune 77.8%)	|2036	|80.7	|74.9	|78	|91.16%|
> |HoloV (prune 77.8%)	|2043	|82.3	|76.5	|79.8	|92.71%|
> |FastV (prune 88.9%)	|1940	|78.6	|69.2	|77.4	|87.96%|
> |HoloV (prune 88.9%)	|2006	|80.7	|72.4	|79.5	|90.52%|
>
> These results demonstrate that HoloV achieves superior performance.

---

> > ### Comment · Reviewer_4HaK · 2025-08-07
> >
> > I sincerely appreciate the authors' detailed response to my concerns — all of my questions have been addressed. I will maintain my score in support of your work.

---

> > > ### Author Response · Authors · 2025-08-07
> > >
> > > Dear Reviewer 4HaK,
> > >
> > > Thank you for your positive assessment and insightful feedback. We are delighted that our responses have addressed all of your concerns, and we greatly appreciate your support for our work.
> > >
> > > Thank you again for your time and thoughtful consideration.
> > >
> > > Best wishes,
> > >
> > > Authors

---

### Official Review · Reviewer_MYQs · 2025-07-09

**Clarity:** 2
**Significance:** 3
**Originality:** 2
**Rating:** 4
**Confidence:** 4

**Summary:**

This paper proposes a new visual token reduction strategy to improve the speed of MLLMs with maintained performance. Compared with prior works that directly reduces tokens with low attention values, the proposed method adaptively distributes the token reduction across spatial crops. This strategy can capture more global visual context, tackling the limitation from prior work that keeps too many semantically similar tokens in the same spatial region. Benchmark results show that the proposed method demonstrates superior efficiency-accuracy trade-off, outperforming the prior SOTA method on various benchmarks.

**Questions:**

See the weaknesses section.

**Ethical Concerns:**

["NO or VERY MINOR ethics concerns only"]

**Final Justification:**

After reading the authors' rebuttal, I think most of my concerns have been addressed. I would like to update my rating to borderline accept.

**Limitations:**

The authors have shown some reasonable limitations in the `Limitations and Future Work` section

**Quality:**

3

**Strengths And Weaknesses:**

Strengths
1. The authors observed that current token pruning method only focuses on the attention value, which could lose the informative global semantics and pay too much attention on the local salient tokens. Thus, this paper proposes a new method to allocate tokens more evenly in a global view, but still consider the local saliency factor. The overall design is reasonable.
2. The benchmark results show that the proposed method achieves SOTA performance on almost all benchmarks over the prior methods. In addition, there is no significant computation/latency overhead of the current method.
3. The paper is relatively easy to understand.

Weaknesses
1. Although the overall idea is relatively easy to understand, some concepts in the paper may need revision:
Line 51 & Line 65: There is a concept called "semantic connectivity". However, it seems there is no evidence showing such "connectivity" between semantics. The goal of the method seems more like to capture more global semantics rather than local features.
Line 150: "Attention dispersion" seems a concept that a small amount of tokens take the majority of the attention. Under this concept, [CLS] attention is steeper in the curve, shows more aligned with the concept of "attention dispersion", but Line 157 - 159 seems to show that last-token attention and equi last attention are more aligned with such concept?
Line 164 - 165: It introduces another concept "global contextual cohesion" which is only mentioned once and never used again. Seems this concept is not necessary.

2. Line 134 - 135 "As shown in Fig. 4 left, adjacent tokens with similar visual features frequently receive comparable attention scores" It seems difficult to reach such conclusion from Fig. 4 left. Please elaborate more on it.

3. Line 131 "we focus on the attention received by visual tokens from the visual [CLS] token" Seems there is a limitation of the only focus on attention from the CLS token. I wonder if the mutual attention between visual token needs to be considered.

4. Sec 4.1 Issues in the formula (1) need to show dimension for the matrices and vectors (2) eq.2 should have $\sum_j$ instead of $\sum$ (3) $\mathcal{H}^c$ is never used (4) $\Omega_c$ is not well explained (5) In eq. 5, why minimizing global diversity $\mathcal{V}^c$? Do we want similar patterns in each crop?

5. Typo: Table 3 Avgerge  -> Average

---

> ### Author Rebuttal · Authors · 2025-07-31
>
> We sincerely thank Reviewer MYQs for the constructive comments on our work. We promise to revise the paper.
>
> ---
>
> > Q1: some concepts in the paper may need revision.
> ﻿
> - Line 51 & Line 65: “Semantic connectivity”
> ﻿
> The concept of  “semantic connectivity” means the cross-region interactions in semantic space, e.g., A on/under/in B, such object relations, retaining only "highlighted tokens" would lose this connectivity. We have clarified in the revision paper.
> ﻿
> To further substantiate this concept, we have added to Figure 3 heatmaps in the revision paper for two representative images (pedestrian and vehicle), showing pairwise attention strengths between adjacent and distant visual tokens.
> ﻿
> - Line 150 & Line 157-159: Inconsistency in “attention dispersion” presentation.
> ﻿
> Sorry for miswriting, "attention dispersion" is a counter concept that more visual tokens take the majority of the attention. Under this concept, compared to [CLS] attention (the top 20% of visual tokens account for 80% of the total attention), text-vision attention tends to be dispersed over more visual tokens, e.g., the top 20% of visual tokens account for only 40% of the total attention, consistent with "attention dispersion". We have revised this mistake in the revision.
> ﻿
> - Line 164 - 165: the concept of "global contextual cohesion" is not necessary.neng
> ﻿
> Thanks for your nice suggestion. We agree, our initial idea means not over-reliance on attention scores, as it disrupts spatial semantic relationships, considering global semantics could mitigate this issue. We have revised this concept in the revision.
>
> ---
>
> > Q2: Line 134 - 135 "As shown in Fig. 4 left, adjacent tokens with similar visual features frequently receive comparable attention scores" It seems difficult to reach such conclusion from Fig. 4 left. Please elaborate more on it.
>
> Take FastV's distribution map of visual token attention as an example, many tokens in regions characterized by flat backgrounds, such as sky, grass, and trees with similar visual features, receive comparable attention scores. When using the top-k strategy to select retained visual tokens, it leads to semantic redundancy, i.e., retaining many visual tokens with semantics of the same objects.
> ﻿
> Further, we added more cases and a statistical analysis to elaborate on this redundancy in the revised Appendix.
>
> ---
>
> > Q3: Line 131 "we focus on the attention received by visual tokens from the visual [CLS] token" Seems there is a limitation of the only focus on attention from the CLS token. I wonder if the mutual attention between visual token needs to be considered.
> ﻿
> We clarified that our primary goal was to study how [CLS] aggregates global visual information, while mutual attention between visual tokens was included in self-attention at ViT, and we don't need this mutual attention between visual tokens to select key visual tokens.
>
> ---
>
> > Q4: Sec 4.1 Issues in the formula (1) need to show dimension for the matrices and vectors (2) eq.2 should have $\sum_j$ instead of $\sum$ (3) $\mathcal{H}^c$ is never used (4) $\Omega_c$ is not well explained (5) In eq. 5, why minimizing global diversity $\mathcal{V}^c$? Do we want similar patterns in each crop?
>
> R(1), we actually have listed dimension of $\mathbf{Z}_v^c \in \mathbb{R}^{M\times d}$, further we follow your comment to show clear dimension, in the formula (1) $\mathbf{S}^c = (\mathbf{1}-\mathbf{I}_M)\odot\mathbf{Z}_v^c {\mathbf{Z}_v^c}^\top$, where $\mathbf{Z}_v^c {\mathbf{Z}_v^c}^\top \in \mathbb{R}^{M\times M}$, and $(\mathbf{1}-\mathbf{I}_M) \in \mathbb{R}^{M\times M}$, thus $\mathbf{S}^c  \in \mathbb{R}^{M\times M}$.
>
> R(2), thanks for your kind suggestion, we have revised from $\sum$ to $\sum_j$.
>
> R(3), $\mathcal{H}^c=\gamma_c \mathcal{V}^c + \mathcal{A}^c$ indicates our holistic attention that is calculated through a balanced scoring mechanism combining contextual diversity and attention saliency, in the code we actually use it to calculate the crop importance weights in formula (4), as there is a writting mistake, we have revised it to $w_c = \frac{(\frac{1}{M}\sum_{t=1}^M \mathcal{H}_t^c)^\tau}{\sum^{\mathcal{C}} (\frac{1}{M} \hat{\mathcal{H}}^{c'})^\tau}$,
>
> where $\hat{\mathcal{H}}^{c'}=\sum_{t=1}^M \mathcal{H}_{t}^{c'}$.
>
> R(4), $\Omega_c$ means the objective has a quota constrain, i.e., $|\Omega_c| = q_c$
>
> R(5), $\mathcal{V}_i^c$ indicates that $i$-th token has diverse connections with others, our objective is to retain those with rich semantics tokens within a crop. To avoid ambiguity, we revised formula (5) to more clearly express the idea of the paper and conform to the actual implementation of the code.
>
> ---
>
> > Q5: Typo: Table 3 Avgerge -> Average
>
> We have revised our paper accordingly, the typo "Avgerge" is revised to "Average" in Table 3.
>
> We sincerely appreciate your valuable suggestions again.

---

> > ### Comment · Reviewer_MYQs · 2025-08-07
> >
> > Thank you for your rebuttal! I think most of my questions are resolved (i.e., Q1, Q2, Q3, Q5). I still have some questions on Q4 (5). In the paper, it is mentioned that "(eq 5) which ensures both crop-wise local saliency and **global relevance**". However, as you explained in the rebuttal, "our objective is to retain those with rich semantics tokens within a crop", which seems not related to ensure global relevance? Could you elaborate more on that? Thanks!
> >
> > Overall, I would like to increase my rating to borderline accept, but still hope the authors to provide more explanation for Q4 (5).

---

> > > ### Author Response · Authors · 2025-08-08
> > >
> > > Dear Reviewer MYQs,
> > >
> > > We sincerely thank you for your time and thoughtful review of our submission. Your constructive feedback has been invaluable in helping us improve the clarity and quality of our work.
> > >
> > > The concept of **global relevance** actually includes the process of adaptive holistic token allocation. We compute a crop-level priority score by averaging token scores within each crop. The total quota for selected image tokens T′ is dynamically allocated to crops according to their normalized crop-level importance. To promote holistic token retention, we forcibly allocate at least one quota to the non-allocated crops, in this way, holistic tokens with different semantics can be considered, and the reservation of locally significant tokens will not be destroyed.
> > >
> > > We hope this clears up your confusion. Thank you again for your time and thoughtful consideration.
> > >
> > > Best wishes,
> > >
> > > Authors

---

> > > > ### Comment · Reviewer_MYQs · 2025-08-08
> > > >
> > > > Thank you for the clarification! Now my concern has been addressed.

---

> ### Author Response · Authors · 2025-08-05
> **Eagerness for Reviewer‘s Valuable Feedback**
>
> Dear Respected Reviewer MYQs,
>
> I hope you are well. With the discussion period ending in less than one day, we wish to confirm all your concerns have been addressed satisfactorily.
>
> Please feel free to share any additional feedback—your insights are invaluable, and we are eager to refine our work accordingly.
>
> Thank you sincerely for your time and effort in reviewing our paper.

---

### Note · Authors · 2025-08-13

We are truly grateful for the invaluable time and detailed feedback provided by all the reviewers.

**It is encouraging to see that Reviewers MYQs, 4HaK, h9EW, cqzx have recognized the significant contributions and positive aspects of our manuscript**, such as the reasonable overall design, superior efficiency-accuracy trade-off (`Reviewer MYQs`); detailed and clear paper, thorough experiments, promising improvements (`Reviewer 4HaK`); solid motivation, exceptional control over performance degradation, well-organized structure (`Reviewer h9EW`); comprehensive experiments across benchmarks, easy integration, substantial computational efficiency gains (`Reviewer cqzx`).

**We have provided detailed responses to all reviewers’ feedback**. Based on the valuable suggestions offered by the reviewers, including clarifying concepts like "semantic connectivity" and "attention dispersion", supplementing experimental results on Qwen2.5-VL and MM-Vet benchmark, revising formulas and fixing typos, adjusting the number of crops dynamically, and integrating new data into the final version. We hope these responses adequately address any potential concerns from the reviewers, the AC, and SPC.

Best regards,

Authors

---

### Decision · Program_Chairs · 2025-09-17

**Decision:**

Accept (poster)

**Comment:**

The paper presents a practical and well-substantiated approach to visual token pruning that improves high-ratio pruning robustness and real-world efficiency. The authors addressed most reviewer concerns with additional experiments and clarifications. Remaining issues (grid-induced fragmentation, heuristic justifications, wider encoder/hardware coverage) are important but do not undermine the main contribution and can be addressed in the camera-ready and future work.

Overall, reviewers converged on accepting or borderline-accepting after the rebuttal, with key conceptual and empirical concerns addressed.